# Can Masked Autoencoders Also Listen to Birds?

**Lukas Rauch**                                      *lukas.rauch@uni-kassel.de*
*University of Kassel*

**René Heinrich**                                    *rene.heinrich@iee.fraunhofer.de*
*Fraunhofer IEE*

**Ilyass Moummad**                                   *ilyass.moummad@inria.fr*
*INRIA Montpellier*

**Alexis Joly**                                      *alexis.joly@inria.fr*
*INRIA Montpellier*

**Bernhard Sick**                                    *b.sick@uni-kassel.de*
*University of Kassel*

**Christoph Scholz**                                 *christoph.scholz@iee.fraunhofer.de*
*Fraunhofer IEE*

**Reviewed on OpenReview:** *https://openreview.net/forum?id=GIBWROXo2J*

## Abstract

Masked Autoencoders (MAEs) learn rich representations in audio classification through an efficient self-supervised reconstruction task. Yet, general-purpose models struggle in fine-grained audio domains such as bird sound classification, which demands distinguishing subtle inter-species differences under high intra-species variability. We show that bridging this domain gap requires full-pipeline adaptation beyond domain-specific pretraining data. Using BirdSet, a large-scale bioacoustic benchmark, we systematically adapt pretraining, fine-tuning, and frozen feature utilization. Our Bird-MAE sets new state-of-the-art results on BirdSet's multi-label classification benchmark. Additionally, we introduce the parameter-efficient prototypical probing, which boosts the utility of frozen MAE features by achieving up to 37 mAP points over linear probes and narrowing the gap to fine-tuning in low-resource settings. Bird-MAE also exhibits strong few-shot generalization with prototypical probes on our newly established few-shot benchmark on BirdSet, underscoring the importance of tailored self-supervised learning pipelines for fine-grained audio domains.

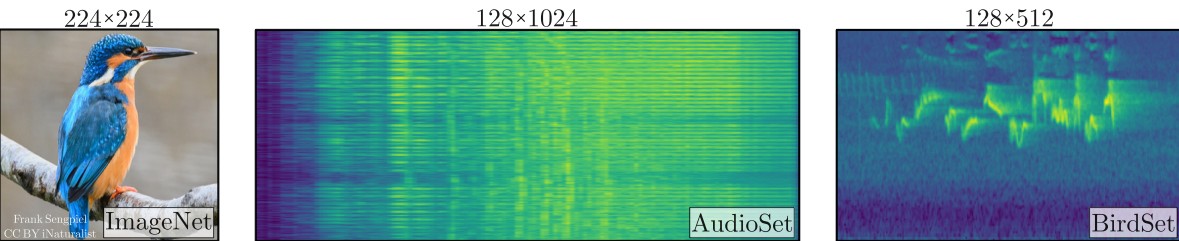

Figure 1: **Input modalities.** Natural images show local spatial correlations. Audio spectrograms exhibit diverse time–frequency patterns. Bird sound spectrograms reveal sparse, harmonic vocal structures.

## 1 Introduction

Representation learning through self-supervised learning (SSL) has emerged as a dominant paradigm in audio classification (Huang et al., 2022; Chen et al., 2023; 2024a), mirroring its impact in computer vision (He et al., 2022; Oquab et al., 2024) and NLP (Touvron et al., 2023). By leveraging vast amounts of unlabeled

data, SSL models learn robust and generalizable representations, often surpassing task-specific supervised models on downstream tasks (Brown et al., 2020). The recent success of masked image modeling (MIM) (He et al., 2022; Chen et al., 2020a) has established it as one of the prevalent SSL pretraining paradigms in vision (Alkin et al., 2025) and audio (Huang et al., 2022). In particular, masked autoencoders (MAEs) (He et al., 2022) efficiently learn rich representations by reconstructing masked inputs, making them scalable for pretraining on large datasets (Bao et al., 2022). However, adapting MAEs from vision to general audio requires addressing the structural properties of spectrograms, such as their distinct local redundancies and time-frequency correlations compared to natural images (cf. Figure 1). This motivated the development of the general-domain Audio-MAE (Huang et al., 2022), pretrained on AudioSet (Gemmeke et al., 2017).

While fine-tuned Audio-MAEs demonstrate competitive performance on audio benchmarks beyond AudioSet, such as ESC-50 (Piczak, 2015), their direct transfer to more fine-grained audio tasks is limited. For instance, general-purpose models exhibit a notable performance gap in specialized tasks such as bird sound classification compared to domain-specific supervised models (Ghani et al., 2023; Hamer et al., 2023). Although AudioSet also contains bird sounds, its coarse-grained nature fails to equip pretrained models with the fine-grained discriminative information required for bird sound classification. In this domain, models must distinguishing between acoustically similar species (low inter-class variation) while handling diverse vocalizations within a single species (high intra-class variation) (Rauch et al., 2024b). Compounding this domain mismatch, MAE's reconstruction objective yields representations that require fine-tuning, offering limited utility as frozen features (Alkin et al., 2025), a drawback common in audio SSL (Chen et al., 2023; 2024a). This presents a critical trade-off: Adapting a general-purpose model via full fine-tuning on different tasks with scarcely labeled data is resource-intensive (Han et al., 2024), while creating a domain-specific foundation model from scratch entails an upfront computational investment. For domains where general audio models seem to plateau (Turian et al., 2022), this one-time investment is justified by the benefits unlocked across the downstream lifecycle. A domain-specific foundation model enables efficient adaptation to various sub-tasks (e.g., population density or call-type classification in bioacoustics) via lightweight probing of frozen representations (Ghani et al., 2023). This efficiency is crucial in fields like bioacoustics, where researchers (e.g., biologists) often face scarce labels and have limited computational resources for edge deployment (Bellafkir et al., 2023).

Given these challenges of fine-grained classification and the need for downstream efficiency, bird sound classification emerges as an ideal testbed for investigating our core hypothesis: achieving state-of-the-art (SOTA) performance in audio currently requires a holistic, domain-aware adaptation of the entire SSL pipeline, including pretraining, fine-tuning and frozen feature utilization. Current SOTA models in this area, such as Perch (Hamer et al., 2023), still rely on supervised learning, failing to capitalize on the vast unlabeled bioacoustic data suitable for SSL. The large-scale BirdSet benchmark (Rauch et al., 2024b), comparable pretraining data volumes to AudioSet, provides the necessary resources to test this hypothesis. Thus, we introduce Bird-MAE, a model pretrained exclusively on bird vocalizations via a fully adapted SSL pipeline. To better leverage its frozen representations, we propose a novel application of prototype learning (Heinrich et al., 2025) as an efficient probing mechanism (i.e., prototypical probing) and systematically evaluate it in low- and high-data regimes. Our key contributions are summarized as follows:

(1) We **empirically demonstrate the domain gap** between a general-purpose model and a domain-specific model. We emphasize the necessity of domain-specific SSL for fine-grained audio tasks.

(2) We **holistically adapt the MAE pipeline for bird sound classification**.[a] We revisit design choices in the pretraining recipe, fine-tuning methods, and utilization of frozen representations.

(3) We introduce **Bird-MAE**[b], a domain-specific MAE trained on BirdSet. It sets new SOTA on BirdSet, improving mAP by 15 percentage points (pp) on average across tasks over prior best models.

(4) We introduce **prototypical probing**, a parameter-efficient probe for frozen features. It lifts mAP by up to 37 pp over linear probes and reduces the fine-tuning gap to 3.3 pp on average across BirdSet.

(5) We establish a **few-shot multi-label benchmark** for BirdSet, where Bird-MAE with prototypical probing approaches full-data performance, highlighting its efficiency in low-data regimes.

---

[a]https://github.com/DBD-research-group/Bird-MAE
[b]https://huggingface.co/collections/DBD-research-group/bird-mae-68a4654a4cc4d363eb87e6f2

**Contributions**

## 2 Related Work

**SSL in audio classification.** SSL in audio classification has advanced the field, spanning from environmental sounds like ESC-50 (Piczak, 2015) to large-scale benchmarks like AudioSet that covers a wide range of sounds (e.g., human, animal, musical). Analogous to ImageNet (Deng et al., 2009) in vision, AudioSet provides a large-scale dataset for pretraining SSL models and evaluating their learned representations. While speech SSL models like Wav2vec2 (Baevski et al., 2020) usually operate on waveforms, general audio classification succeeds by adapting vision-based SSL techniques to spectrograms, achieving SOTA results on AudioSet (Chen et al., 2023; 2024a). MIM has successfully transitioned to audio classification by reconstructing masked spectrogram patches, introducing the Audio-MAE (Huang et al., 2022). This pretraining paradigm offers computational efficiency and fosters the learning of rich audio representations from unlabeled data. Subsequent work, such as BEATs (Chen et al., 2023) and EAT (Chen et al., 2024a), further progress audio MIM, incorporating a teacher-student approach. Our work centers on the MAE architecture. Its conceptual simplicity and pretraining efficiency make it an ideal baseline for isolating the impact of domain-specific adaptation and assessing the efficacy of novel probing techniques in challenging fine-grained domains.

**Transfer learning in audio classification.** General-purpose audio SSL models have proven effective for diverse downstream tasks (Turian et al., 2022; Saeed et al., 2021). BEATs (Chen et al., 2023), EAT (Chen et al., 2024a) and Audio-MAE (Huang et al., 2022) demonstrate their performance on tasks such as speech emotion recognition or environmental sound classification. However, recent studies reveal a notable performance degradation when these general-purpose models are applied to highly specialized domains, particularly bioacoustics (Hamer et al., 2023; Ghani et al., 2023). Benchmarks designed for transfer learning, such as HEAR (Turian et al., 2022), and bioacoustic benchmarks like BirdSet (Rauch et al., 2024b) or BIRB (Hamer et al., 2023) highlight this limitation. Specifically, Audio-MAE (Huang et al., 2022) performs worse than spectrogram-based features from supervised models in bioacoustic tasks (Ghani et al., 2023). This performance drop underscores the current limitations of relying on general-domain pretraining for more fine-grained audio tasks and motivates the development of domain-specific solutions. In this work, we address this limitation by introducing and evaluating Bird-MAE specifically adapted to bird sound classification, quantifying the benefits of domain-specific solutions for fine-grained classification in audio.

**Downstream task adaptation in SSL.** Adapting models to downstream tasks typically involves full fine-tuning or utilizing frozen representations with lightweight probes (Marks et al., 2025). While fine-tuning often yields the highest performance, it can be computationally expensive and may lead to overfitting on smaller datasets. Thus, utilizing frozen representations has gained notable interest in vision (Oquab et al., 2024; El-Nouby et al., 2024; Xie et al., 2022; Assran et al., 2023). Standard approaches involve extracting features (e.g., via global average pooling or the CLS-token) and training a simple classifier, such as linear probing (Oquab et al., 2024), k-NN probing (Zhou et al., 2021; Kakogeorgiou et al., 2022; Lehner et al., 2024), or shallow MLP probing (Dubois et al., 2022; Fuller et al., 2023; Tschannen et al., 2023). However, it is widely observed that representations learned via generative tasks like MIM underperform with linear probing compared to contrastive methods (Alkin et al., 2025; He et al., 2022; Park et al., 2023). To mitigate this gap in MIM, upstream feature refinement (Alkin et al., 2025) or alternative probing methods have been explored in vision. Notably, attentive probing (El-Nouby et al., 2024; Lee et al., 2019) applies an attention mechanism over patch tokens and improves frozen representations with low computational overhead (Yu et al., 2022; Chen et al., 2024b; Darcet et al., 2025). Another probing paradigm involves prototypical networks (Snell et al., 2017; Palanisamy et al., 2024; Tian et al., 2024), which extract class centroids from frozen representations for a similarity-based class assignment without retraining. Despite these advancements in vision, current best-performing audio SSL models (Huang et al., 2022; Chen et al., 2023; 2024a) rely on full-model fine-tuning, suggesting their frozen representations offer suboptimal performance or are not properly utilized. In our setting, we evaluate standard parametric probing methods for downstream adaptation using frozen representations, including linear, MLP, and attentive probing. Additionally, we propose and analyze prototypical probing, utilized from prototypical networks in bioacoustics (Heinrich et al., 2025) for a new purpose: as a lightweight, parameter-efficient probe for frozen MAE representations, effectively utilizing their spatial features.

**Bird sound classification.** Supervised learning has dominated research in bird sound classification, typically employing convolutional architectures that remain the top performers on bioacoustic benchmarks (Stowell,

2021; Rauch et al., 2024b). For instance, Google's Perch model (Hamer et al., 2023) is based on the EfficientNet architecture (Tan & Le, 2019). The feasibility of large-scale supervised training stems from community-driven platforms like Xeno-Canto (XC) (Vellinga & Planqué, 2015), which currently hosts over 850k weakly-labeled bird recordings. Perch and BirdNeXt (Rauch et al., 2024b) derive their training data from XC. However, reliance on manually curated, non-standardized datasets from these platforms has hindered comparison across studies and methods (Rauch et al., 2024b). The introduction of the BirdSet dataset and multi-label bird classification benchmark, which contains volumes of pretraining data comparable to AudioSet, makes this comparison possible for domain-specific SSL. While SSL has shown promise in speech and general audio classification, its evaluation in bioacoustics is less mature. NatureLM-audio (Robinson et al., 2025) presents the first audio-language model tailored to general bioacoustics, demonstrating competitive zero-shot performance on BirdSet. Existing domain-specific SSL models for bioacoustics, such as BirdAVES (Hagiwara, 2023) and contrastive models (Moummad et al., 2024), have not yet been evaluated under the standardized conditions provided by BirdSet. This work introduces the domain-specific Bird-MAE and comprehensively evaluates it on BirdSet, establishing a new SOTA.

## 3 Model and Training Methodology for Domain-Specification

This section details the methodological modifications applied to the baseline Audio-MAE architecture and training procedure as part of our holistic domain specification to bird sounds. We organize these modifications into three modules. First, the *pretraining module* (M1) outlines our changes to the pretraining recipe of the baseline Audio-MAE. Second, we introduce two modules addressing the main downstream adaptation strategies for a pretrained SSL model. The *fine-tuning module* (M2) involves modifications for the full model training process, while the *frozen representations module* (M3) investigates the pretrained model as a fixed feature extractor. In the following, we motivate and detail the modifications within each module.

### 3.1 Pretraining (M1)

Pretraining lays the foundation for effective SSL by learning representations adaptable to downstream tasks. The core of the Audio-MAE baseline is the pretrained encoder $h_\alpha : \mathcal{X} \subseteq \mathbb{R}^{F \times T} \to \mathbb{R}^{H \times W \times D}$ with parameters $\alpha$ from AudioSet. Here, $F$ represents the number of frequency bins and $T$ the number of time frames of an input spectrogram $\mathbf{x} \in \mathcal{X}$. The encoder mAPs $\mathbf{x}$ to a patch-based feature map $\mathbf{h}_\alpha(\mathbf{x})$, where $H$ and $W$ denote the number of non-overlapping patches along the height and width dimensions, and $D$ is the feature dimension per patch. For instance, given an AudioSet spectrogram image with $F = 128$ and $T = 1024$ with a patch size of $16 \times 16$, the encoder $h_\alpha$ produces a feature map $\mathbf{h}_\alpha(\mathbf{x})$ of dimension $8 \times 64 \times D$. Our proposed modifications result in an encoder $h$, pretrained on bird sounds, denoted $h_\beta$. The key changes from the baseline Audio-MAE pretraining from Huang et al. (2022) are listed in Table 1 with the detailed ablations in Section 5.2. The modifications within this module include:

| Model | Dataset | Img | Decoder | Epochs | Masking | Batch | Mean | Std | Mixup | LR | WD |
|-------|---------|-----|---------|--------|---------|-------|------|-----|-------|-----|-----|
| **Audio-MAE** | AS-2M | 1024×128 | Swin | 32 | 0.8 | 512 | -4.2 | 4.569 | 0 | 2e-4 | 1e-4 |
| **Bird-MAE** | XCL-1.7M | 512×128 | ViT | 150 | 0.75 | 1024 | -7.2 | 4.43 | 0.3 | 2e-4 | 1e-4 |

Table 1: **Comparison of pretraining parameters** of Audio-MAE (baseline) and Bird-MAE (our model).

**Data source.** The choice of pretraining data influences downstream performance, especially when adapting models from coarse-grained to fine-grained classification tasks. General-purpose datasets like AudioSet encompass a broad spectrum of acoustically distinct classes (high inter-class variation), encouraging models to learn discriminative general features. However, fine-grained domains like bird sound classification require distinguishing between subtle different species (low inter-class variation) while handling acoustic variability within each species (high intra-class variation) (Rauch et al., 2024b). AudioSet, despite including some animal sounds, does not adequately prepare models for these fine-grained bioacoustic nuances. Therefore, to develop a model tailored for this challenge, we replace AudioSet with domain-specific pretraining data derived from BirdSet (`XCL-1.7M` after curation, see Section 4).

**Data processing.** The raw pretraining dataset from BirdSet contains over 3 million event samples. However, the raw audio collection suffers from redundancy (e.g., multiple events per file, similar background noise) and class imbalance, which can degrade SSL performance (Balestriero et al., 2023). Inspired by the curation process from Oquab et al. (2024), we apply a small selection procedure based on available metadata to reduce redundancy. Specifically, we limit the maximum number of event samples retained per species and recording file. This process results in our curated pretraining dataset `XCL-1.7M`, approximately halving the original size of 3.4 million vocalization events in BirdSet. This curated set is used to train the encoder $h_\beta$. Further details on the curation process and the results are provided in Section 5 and the Appendix A.

**Training parameters.** Optimizing the pretraining recipe can yield substantial performance gains in SSL (Oquab et al., 2024). Thus, we systematically examine the training recipe of the Audio-MAE baseline and identify modifications that yield improvements for bird sound classification through model optimization. These include adjustments to the decoder architectures, increasing the number of training epochs, refining the masking ratio, increasing the batch size, and incorporating mixup augmentation (Zhang et al., 2018) during pretraining. These key changes are summarized in Table 1, with detailed ablation studies in Section 5.

### 3.2 Fine-tuning (M2)

During downstream adaptation through fine-tuning, the baseline Audio-MAE applies average pooling to the encoder's output feature map $\mathbf{h}_\alpha(\mathbf{x})$ to obtain a compact embedding $\bar{\mathbf{h}}_\alpha(\mathbf{x}) \in \mathbb{R}^D$. This embedding is then fed into a linear classification head $f_\psi : \mathbb{R}^D \to \mathbb{R}^C$ (with parameters $\psi$ and $C$ classes) trained along the encoder to produce logits $\mathbf{z} = f_\psi(\bar{\mathbf{h}}_\alpha(\mathbf{x}))$. This module details modifications applied during this process.

**Domain augmentations.** To bridge the inherent domain shift between training and test data in BirdSet[1], we utilize domain-specific augmentations while fine-tuning the encoder. While Audio-MAE also employs augmentations, it does not apply strategies tailored to bioacoustics. Our augmentations, informed by results from Rauch et al. (2024b), are designed to simulate common acoustic variations in bird recordings, such as diverse background noises (i.e., noise mixing), varying signal strengths (i.e., gain mixing), and the co-occurrence of multiple vocalizations (i.e., mixup). We supplement these with spectrogram-level augmentations, including frequency and time masking. Further details on the augmentations are provided in Table 9.

**Prototypical pooling.** Inspired by the performance improvements of the supervised AudioProtoPNet (Heinrich et al., 2025; Chen et al., 2019; Donnelly et al., 2022) in bioacoustics, we introduce a prototypical pooling layer $f_\phi$. The prototypical layer explicitly pools the spatial structure of the pretrained encoder's patch embeddings $\mathbf{h}_\beta(x) \in \mathbb{R}^{H \times W \times D}$, comparable to attentive pooling (El-Nouby et al., 2024). For each class $c \in \{1, \ldots, C\}$, we learn a set of $J$ class-specific prototype vectors $\{\mathbf{p}_{c,j}\}_{j=1}^J$, with each prototype $\mathbf{p}_{c,j} \in \mathbb{R}^D$. These prototypes are randomly initialized as learnable parameters, distinct from the encoder's weights. We then compute the cosine similarity scores between each prototype $\mathbf{p}_{c,j}$ and every patch embedding $\mathbf{h}_\beta(\mathbf{x})_{h,w}$. The resulting similarity scores are then aggregated via max-pooling across all spatial dimensions to obtain the highest similarity score for each prototype of class $c$:

$$\bar{s}_{c,j} = \max_{h,w} \frac{\mathbf{p}_{c,j} \cdot \mathbf{h}_\beta(x)_{h,w}}{\|\mathbf{p}_{c,j}\|\|\mathbf{h}_\beta(x)_{h,w}\|}, \quad h = 1, \ldots, H; \ w = 1, \ldots, W. \tag{1}$$

This yields $J$ similarity scores $(\bar{s}_{c,1}, \ldots, \bar{s}_{c,J})$ for each class $c$. The prototypical layer $f_\phi$ then transforms these class-specific similarity scores into logits. Following Heinrich et al. (2025), this transformation is implemented for each class $c$ by a dedicated linear layer, $g_c : \mathbb{R}^J \to \mathbb{R}$, which takes the $J$ similarity scores for that class as input to produce a single scalar logit $\bar{z}_c$. Each $g_c$ uses weights that are constrained to be non-negative, ensuring that a higher similarity to a class prototype contributes positively to the class logit. We adopt the initialization from Heinrich et al. (2025): weights are set to 1 for uniform initial prototype weighting and biases to -2. This yields a near-zero sigmoid probability for instances with no similarity to the prototypes of a class, which is suitable for multi-label classification. The final logit vector for all classes is formed by concatenating these individual class logits. This design ensures that the prediction for each class is based solely on the evidence from its associated prototypes, leveraging local spatial features for robust classification.

---

[1]BirdSet training data consists of focal (directed) recordings, contrasting to test data from omnidirectional soundscapes.

To encourage diversity among the learned prototypes within each class and prevent redundancy, the overall training loss incorporates an equally weighted orthogonality loss term, adapted from Donnelly et al. (2022).

### 3.3 Frozen Representations (M3)

Frozen representations offer a computationally efficient alternative to full fine-tuning (Oquab et al., 2024; Touvron et al., 2023). However, MIM representations primarily capture reconstruction-oriented patterns rather than discriminative features (Alkin et al., 2025; Oquab et al., 2024). This limits their direct usability, as the task may dilute critical classification features across reconstructed regions (Walmer et al., 2023). While fully fine-tuning the encoder $h$ typically addresses this issue (He et al., 2022; Park et al., 2023), it can be computationally expensive and unsuitable for tasks with little available labeled data. Thus, this module investigates probing techniques to leverage frozen representations from the pretrained MAE encoder $h_\beta$.

**Prototypical probing.** We propose prototypical probing as a parameter-efficient method to leverage frozen features. This involves adapting the prototypical pooling layer $\mathbf{f}_\phi$ (as described in M2) to the frozen encoder $h_\beta$ as a lightweight probing head. It is a parametric method: The parameters of the prototype vectors $\{\mathbf{p}_{c,j}\}_{j=1}^J$ for all classes $c$ and the final class-specific linear layers $\{g_c\}_{c=1}^C$ are trained. Similar to attentive pooling, prototypical probing utilizes the full spatial feature map $\mathbf{h}_\beta(x) \in \mathbb{R}^{H \times W \times D}$, preserving local structural information. This might be beneficial for bird sounds, as vocalizations typically occupy small regions of the spectrogram, where global averaging may dilute this information. Additionally, prototypical probing is parameter-efficient as the additional trainable parameters only consist of the prototypes $J \cdot C \cdot D$ and the final linear layer with a total of $J \cdot C + C$ parameters. While this scales linearly with the number of classes and prototypes, its total size remains negligible compared to the encoder. For instance, with the ViT-L encoder (approx. 300M parameters), prototypical probing for the `HSN` task adds only about 430k parameters (Table 5). Furthermore, it is often smaller than the overhead of attentive probing, which adds approximately $2D^2 + D$ parameters (El-Nouby et al., 2024). Prototypical probing retains the low parameter characteristic of linear probing while efficiently exploiting non-linear spatial information crucial for discriminative performance with frozen MAE features.

## 4 Data and Processing

**Dataset.** Our experiments utilize BirdSet (Rauch et al., 2024b), a comprehensive benchmark for multi-label bird sound classification (i.e., classifying bird species based on their vocalizations). Unlike AudioSet, where each 10-second sample captures a wide array of sounds, bird calls are typically shorter (within 5 seconds, comparable to ESC-50 (Piczak, 2015)) and are confined to narrow frequency bands. BirdSet aggregates the training data from XC (Vellinga & Planqué, 2015), encompassing approximately 520,000 unique recordings

| Dataset | | \|**Train**\| Recordings | \|**Train**\| Events | \|**Test**\| Segments | #**Classes** |
|---|---|---|---|---|---|
| *Pretraining* | | | | | |
| Xeno-Canto Large | XCL | 528,434 | 1,724,598 | - | 9,735 |
| *Downstream Tasks* | | | | | |
| High Sierra Nevada | HSN$_{\text{val}}$ | 5,460 | 17,938 | 12,000 | 21 |
| Powdermill Nature | POW | 14,911 | 2,586 | 4,560 | 48 |
| Amazon Basin | PER | 16,802 | 5,743 | 15,120 | 132 |
| Colombia Costa Rica | NES | 16,117 | 4,034 | 24,480 | 89 |
| Hawaiian Islands | UHH | 3,626 | 12,978 | 36,637 | 27 |
| France and Spain | NBP | 24,327 | 76,438 | 563 | 51 |
| Sapsucker Woods | SSW | 28,403 | 4,285 | 205,200 | 81 |
| Sierra Nevada | SNE | 19,390 | 2,557 | 23,756 | 56 |

Table 2: **Dataset overview of BirdSet for pretraining and downstream tasks**. \|**Train**\| Recordings contains the number of recordings, \|**Train**\| Segments is the number of extracted samples per task in our experiments, and \|**Test**\| is the number of 5-second segments. `HSN`$_{\text{val}}$ is used for validation and ablations.

(weakly labeled at the file level) from nearly 10,000 bird species, totaling over 3 million vocalization events. For evaluation, BirdSet provides eight downstream tasks, each consisting of a dedicated training subset and a test set derived from fully annotated soundscape recordings from different geographical regions (e.g., High Sierras Nevada (HSN) or Amazon Basin (PER)). The test sets are segmented into 5-second intervals, where each interval receives multi-label annotations indicating the presence (one or multiple) or absence of birds. This structure explicitly captures challenges like domain shift between training (focal) and test (soundscape) data, as detailed in Rauch et al. (2024b). Table 2 provides a detailed overview of the datasets.

**Processing and evaluation.** Audio segments are standardized to 5 seconds and sampled to 32 kHz. We extract 128-dimensional log-mel filterbank features, following common practice in audio SSL (Huang et al., 2022; Chen et al., 2024a). The resulting input dimensions are fixed at $128 \times 512$. All results in ablations and benchmark studies are averaged over three repetitions. We report the class-based mean average precision (mAP). Since BirdSet provides no typical validation split per downstream task for dedicated training, we repurpose the HSN downstream task (with only 21 classes) as our development set for hyperparameter tuning and ablation studies. After finalizing all design choices, we retrain the model once on each downstream task's training data and report performance on their test sets.

## 5 Ablation Studies

This section presents ablation studies to validate the design choices and quantify the impact of each modification module (M1, M2, M3) introduced in Section 3. We analyze these components by sequentially applying them to a baseline Audio-MAE configuration, which uses the original implementation from Huang et al. (2022) detailed in Table 1. We illustrate the cumulative improvements with full *fine-tuning* in Figure 3.

**Settings.** For modifications related to the pretraining (M1, Section 5.1) and fine-tuning (M2, Section 5.2) modules, we evaluate performance via full model *fine-tuning*. For frozen representations (M3, Section 5.3), we ablate the effectiveness of different *probing techniques* (linear, MLP, attentive), compared to our prototypical probing when using the best-performing settings from M1 and M2. For each experiment, we report the average over three random seeds to account for variability in training. All ablation experiments are performed on the HSN multi-label downstream task from BirdSet. Further experimental details and hyperparameters are provided in the Appendix D.

### 5.1 Pretraining (M1)

**Data source.** Figure 3 shows that replacing the AudioSet pretraining data with BirdSet for the base model yields a modest performance gain of 2.45 pp in fine-tuning. Even if extensive fine-tuning on large downstream datasets can mitigate pretraining domain mismatch, domain-specific pretraining data provides a clear advantage, especially for the performance of probing techniques (see Section 5.3). However, the most decisive gains emerge from a holistic adaptation of the entire SSL pipeline to the target domain. Thus, swapping in domain-specific data is necessary but insufficient: aligning both objective and downstream training with the structure of bird vocalizations in the domain unlocks most of the benefit.

**Data processing.** The quality and size of the pretraining dataset are crucial for SSL. To quantify the impact of dataset size, we begin with the full XCL-3.4M_R training dataset with all available sound events and progressively reduce it to 50%, 25%, and 12.5% by random sampling. As shown in Table 3a, pretraining performance generally scales with dataset size when using randomly sampled subsets, although gains diminish beyond 1.7 million samples in our use case. Noticeably, applying our data curation strategy (balancing classes, reducing redundancy via metadata) to create the curated 1.7 million sample dataset (XCL-1.7M) results in better performance compared to using an uncurated dataset of the same size or even the full, uncurated 3.4 million sample dataset. This highlights the benefit of data curation, outweighing data volume in this setting. More details of the curation process are available in the Appendix A.

**Pretraining recipe.** Optimizing the pretraining recipe beyond just the data source further enhances performance. As summarized in Figure 3, modifications like increased epochs, adjusted masking ratio, larger batch size, and mixup (see Table 3b) sequentially improve downstream results. While changing the decoder architecture did not yield direct performance gains in isolation, it improved training stability. Figure 2

| Dataset | Selection | MAP |
|---------|-----------|-----|
| XCL-3.4M$_R$ | Random | 52.18 |
| XCL-1.7M$_R$ | Random | 52.11 |
| XCL-0.8M$_R$ | Random | 43.28 |
| XCL-0.4M$_R$ | Random | 35.98 |
| **XCL-1.7M** | **Curated** | **55.28** |

(a) Dataset size

| Mix | MAP |
|-----|-----|
| 0.0 | 53.65 |
| **0.3** | **55.28** |
| 0.5 | 52.56 |
| 0.7 | 52.39 |

(b) Mixup

| Pool | MAP |
|------|-----|
| cls | 52.89 |
| avg | 53.15 |
| atten | 53.36 |
| **proto** | **55.28** |

(c) Pooling

Table 3: **Detailed ablations** on (a) dataset size and curation in SSL, (b) mixup in SSL and (c) pooling in fine-tuning.

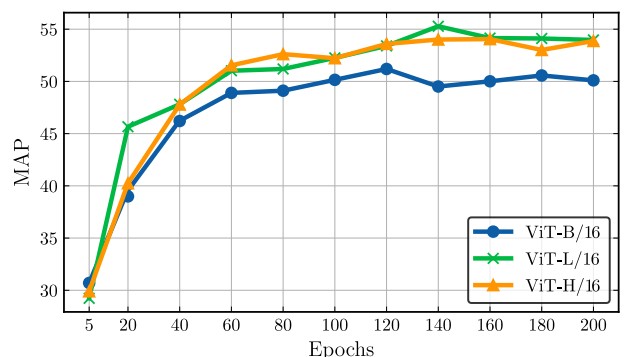

Figure 2: **Model size and training epochs comparison** on `HSN`. We report the MAP score at different pretraining checkpoints in all model sizes.

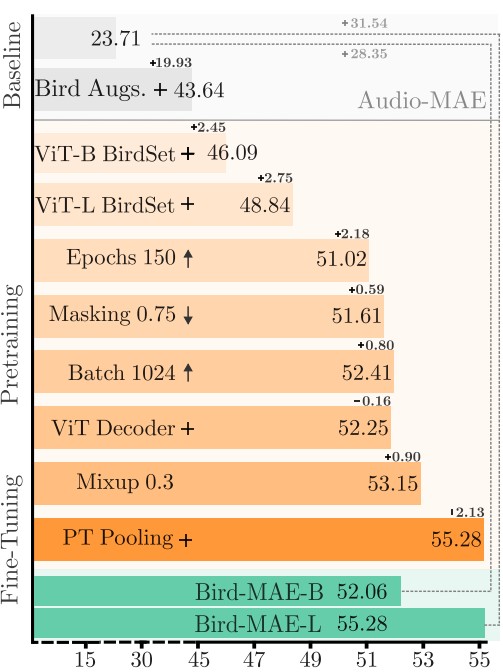

Figure 3: **Ablations for improving the base Audio-MAE**. The MAP results are reported on `HSN`. The + symbol indicates a new component, while ↑ and ↓ denote an increase and a decrease in a parameter.

confirms the benefit of extended pretraining, showing mAP improvements across different ViT sizes up to approximately 150 epochs, after which gains saturate.

## 5.2 Fine-tuning (M2)

**Domain augmentations.** Adapting the fine-tuning process with domain-specific data augmentations is crucial. Our sequential ablation in Figure 3 shows that applying the baseline Audio-MAE to `HSN` without domain-adapted augmentations yields inferior results (23.71%). Introducing domain-specific augmentations (detailed in the Appendix D) during fine-tuning provides a substantial performance uplift, increasing mAP by circa 29 pp over the baseline implementation. This highlights the importance of domain-aware adaptations, ensuring models can deal with the challenges in the domain data (e.g., domain shift in BirdSet).

**Prototypical pooling.** Replacing global averaging or the CLS token with prototypical pooling further enhances classification performance when fully fine-tuning the model, setting new SOTA results. For the Bird-MAE-L model on `HSN`, this final modification elevates the mAP score to 55.28% from 53.15% when using global average pooling. Prototypical pooling also outperforms alternative advanced pooling mechanisms such as attentive pooling (see Table 3c). This improvement contributes to more than doubling the performance of the initial Audio-MAE baseline (23.71% mAP, before any domain specifications). As shown in Table 5, the parameter overhead of this prototypical pooling operation is minimal compared to the ViT encoder.

## 5.3 Frozen Representations (M3)

**Prototypical probing.** We ablate the quality of frozen representations using various probing methods in Table 4. We use the best-performing settings from M1 and M2, including all augmentations. All experiments use $J$=20 prototypes. The impact of varying $J$ is shown in Appendix F. Consistent with findings in MIM (Alkin et al., 2025), our results confirm that standard probing techniques applied to features extracted

via global average pooling (linear, MLP) perform poorly with frozen MAE representations, even with the domain-specific Bird-MAE. mAP scores remain notably lower than full fine-tuning. However, methods explicitly leveraging the spatial feature map achieve considerable performance gains. Attentive probing notably improves results across model sizes. Our prototypical probing further boosts performance, outperforming attentive probing across all Bird-MAE model sizes while remaining more parameter-efficient (Table 5) with approximately 20% of parameters compared to attentive probing for this dataset. Interestingly, prototypical probing performs worse than attentive probing when applied to the general-purpose Audio-MAE, highlighting that prototype-based methods might benefit from domain-specific pretraining. For the Bird-MAE-L model, prototypical probing achieves a mAP of 49.97%, a substantial gain from MLP probing (+34.75 pp) and only 5.31 pp below the full fine-tuning result (55.28%). This demonstrates that by effectively utilizing the spatial information preserved in the frozen MAE feature map, prototypical probing addresses the limitations of frozen MIM representations for discriminative tasks. It offers an efficient alternative to full fine-tuning.

| ViT | Technique | Linear | MLP | Attentive | Prototypical |
|-----|-----------|--------|-----|-----------|--------------|
| | | Fine-tuning: **43.11** | | | |
| B/16 | Audio-MAE | 9.35 | 10.45 | **31.43** | 20.89 |
| | | Fine-tuning: **52.06** | | | |
| B/16 | Bird-MAE | 13.06 | 17.23 | 43.12 | **43.84** |
| | | Fine-tuning: **55.28** | | | |
| L/16 | Bird-MAE | 13.29 | 15.22 | 47.81 | **49.97** |
| | | Fine-tuning: **54.05** | | | |
| H/16 | Bird-MAE | 13.83 | 18.71 | 45.73 | **47.52** |

Table 4: **Frozen representation ablations** on HSN, evaluated with probing techniques. Linear and MLP utilize the global average, attentive and prototypical ($J = 20$) the feature map.

| Probing | Parameters | HSN |
|---------|-----------|-----|
| | **ViT-L parameters: 307M** | |
| Linear | $C(D+1)$ | 21k |
| MLP | $H(D+1) + C(H+1)$ | 535k |
| Attentive | $2D^2 + (C+1)D + C$ | 2.1M |
| Prototypical | $C \cdot [J(D+1) + 1]$ | 430k |

Table 5: **Parameters for probing** with example values of HSN: $D = 1024$, $C = 21$, $H = 512$, and $J = 20$.

# 6 Benchmark Results

This section presents the empirical evaluation of our domain-specific Bird-MAE model on the BirdSet downstream tasks, comprising multi-label classification of bird species vocalizations. We assess performance under two conditions: BirdSet's *multi-label classification* using all available training data and our novel *few-shot multi-label probing* benchmark with limited labeled examples. Our evaluation aims to (1) showcase the importance of a domain-specific SSL in audio, (2) validate the effectiveness of prototypical probing for leveraging frozen representations, and (3) establish new SOTA results on BirdSet.

**Baselines and evaluation.** We compare Bird-MAE against several relevant models. Our baseline is the Audio-MAE-Base[2] pretrained on AudioSet and fine-tuned with prototypical pooling. First, we compare against other bird-specific SSL models: BirdAVES (Hagiwara, 2023) and a SimCLR (Chen et al., 2020b) implementation from Moummad et al. (2024), both pretrained on custom XC data. Second, we include results from the best-performing supervised models in bird sound classification: Google's Perch (Hamer et al., 2023) and BirdSet's BirdNeXt (Rauch et al., 2024b), also pretrained on XC data. We report Bird-MAE results using ViT-Base, ViT-Large, and ViT-Huge backbones, incorporating all modifications from Section 3. Our domain-specific augmentation pipeline is applied during fine-tuning and probing to all models where feasible to ensure fair comparison in both tasks. We exclude spectrogram-level augmentations for the waveform-based BirdAVES model. For Perch[3] and BirdNeXt, we mask logits for classes not present in the downstream task's label set (Rauch et al., 2024b). Hyperparameters for all models are tuned once on HSN before final evaluation across downstream tasks. More details and hyperparameters can be found in Appendix D.

---

[2]Larger model checkpoints are not available from the source paper (Huang et al., 2022).
[3]Perch is not publicly available for fine-tuning.

| Model | Arch. | Pretraining | HSN$_{val}$ | POW | PER | NES | UHH | NBP | SSW | SNE |
|---|---|---|---|---|---|---|---|---|---|---|
| *Supervised (with masked inference)* | | | | | | | | | | |
| Perch | EffNet-B1 | Xeno-Canto* | 41.84 | 30.41$_{val}$ | 18.23 | 38.53 | 26.71 | 62.62 | 28.11 | 28.45 |
| BirdNeXt | ConvNeXt | XCL BirdSet | 47.29 | 35.63$_{val}$ | 18.52 | 33.68 | 26.07 | 62.24 | 35.19 | 29.88 |
| *Self-Supervised (with fine-tuning)* | | | | | | | | | | |
| BirdAVES | HuBERT | Xeno-Canto* | 42.63 | 11.33 | 5.18 | 5.72 | 21.97 | 70.09 | 4.45 | 7.08 |
| SimCLR | CvT-13 | Xeno-Canto* | 39.33 | 34.84 | 16.85 | 23.54 | 20.72 | 65.69 | 18.67 | 15.99 |
| Audio-MAE | ViT-B/16 | AS-2M | 44.69 | 39.03 | 21.32 | 29.83 | 26.43 | 67.02 | 26.94 | 22.44 |
| Bird-MAE | ViT-B/16 | XCL-1.7M | 52.06 | 45.24 | 27.58 | 37.12 | 28.48 | 62.86 | 32.83 | 28.04 |
| | ViT-L/16 | XCL-1.7M | **55.28** | **55.26** | **34.64** | **41.50** | **30.17** | **71.69** | 40.82 | **33.82** |
| | ViT-H/16 | XCL-1.7M | 54.80 | 54.05 | 33.29 | 39.28 | 29.81 | 69.35 | **41.32** | 32.18 |

Table 6: **Fine-tuning results on the multi-label classification benchmark with full data (mAP%).** Comparison of SL and SSL models, following the evaluation protocol of BirdSet. **Best** and second best results are highlighted. Xeno-Canto* denotes pretraining on unspecified subsets of XC data.

## 6.1 Multi-Label Classification

**Settings.** In this section, we first evaluate the performance of the pretrained Bird-MAE on BirdSet's multi-label classification benchmark with *full training data*. We present the fine-tuning results in Table 6 and the frozen representation results in Table 7. For each experiment, we report the mean over three randomly initialized runs. More detailed results are available in Section E.1.

**What is the performance gain of a domain-specific MAE?** Table 6 confirms results from our ablation studies: Domain specification via Bird-MAE yields substantial performance improvements across the BirdSet benchmark compared to the general-purpose Audio-MAE. While the Bird-MAE-B also offers notable gains over the available Audio-MAE baseline, the benefits become more pronounced with larger architectures. For instance, our Bird-MAE-L achieves notably higher mAP scores than the baseline, with performance gains of +15 pp on POW (+7 pp Bird-MAE-B) or +13 pp on PER (+6 pp Bird-MAE-B). Furthermore, Bird-MAE consistently and notably outperforms the other domain-specific SSL baselines AVES and SimCLR across all datasets, often by margins exceeding 15-20 pp mAP on average.

**How does the model compare to supervised models?** We compare Bird-MAE against the current best-performing supervised models (Perch and BirdNeXt). Our fine-tuned Bird-MAE models consistently achieve new SOTA results across the BirdSet benchmark. Specifically, Bird-MAE-L outperforms the BirdNeXt and Perch baselines on all eight datasets, often by considerable margins. For example, on SSW, Bird-MAE-L achieves 40.82% mAP compared to Perch's 28.11% mAP (+12.7 pp), and on PER, it achieves 34.64% mAP versus Perch's 18.23% mAP (+16.4 pp). These results demonstrate the effectiveness of domain-specific SSL combined with modern transformer architectures and higher parameter counts compared to prior supervised CNN-based approaches in bird sound classification.

**Can we freeze the representations?** While audio SSL currently relies on fine-tuning (Chen et al., 2023; 2024a), efficient deployment is highly desirable for edge applications in bioacoustics (Höchst et al., 2022). We evaluate the performance of frozen representations using linear versus our proposed prototypical probing in Table 7. Linear probing performs poorly for Audio-MAE and Bird-MAE across backbone sizes, confirming the difficulty of using frozen MIM representations directly. However, prototypical probing drastically improves performance by leveraging the spatial feature map: it closes the gap to fine-tuning to approximately 3 pp mAP on average across downstream tasks. Bird-MAE with prototypical probing also achieves substantial gains over Audio-MAE with prototypical probing (e.g., +27.3 pp base performance on NBP). Additionally, it notably outperforms other SSL models (AVES, SimCLR) using either probing method and also surpasses the fully supervised Perch model (from Table 6) on nearly all datasets using only frozen representations. This challenges the notion that MAE features are unsuitable for probing and demonstrates prototypical probing as a highly effective and efficient alternative to fine-tuning in bird sound classification. While prototypical

probing enhances performance for the masking-based BirdAVES model, it seems to degrade performance for the contrastive SimCLR model, suggesting probing effectiveness interacts with the SSL pretraining objective.

| Model | Arch. | Probing | $HSN_{val}$ | POW | PER | NES | UHH | NBP | SSW | SNE |
|-------|-------|---------|------|-----|-----|-----|-----|-----|-----|-----|
| *Self-Supervised (with frozen representations)* | | | | | | | | | | |
| BirdAVES | HuBERT | linear | 14.91 | 12.60 | 5.41 | 6.36 | 11.76 | 33.68 | 4.55 | 7.86 |
| | | proto | 32.52 | 19.98 | 5.14 | 11.87 | 15.41 | 39.85 | 7.71 | 9.59 |
| SimCLR | CvT-13 | linear | 17.29 | 17.89 | 6.66 | 10.64 | 7.43 | 26.35 | 6.99 | 8.92 |
| | | proto | 18.00 | 17.02 | 3.37 | 7.91 | 7.08 | 26.60 | 5.36 | 8.83 |
| Audio-MAE | ViT-B/16 | linear | 8.77 | 10.36 | 3.72 | 4.48 | 10.78 | 24.70 | 2.50 | 5.60 |
| | | proto | 19.42 | 19.58 | 9.34 | 15.53 | 16.84 | 35.32 | 8.81 | 12.34 |
| Bird-MAE | ViT-B/16 | linear | 13.06 | 14.28 | 5.63 | 8.16 | 14.75 | 34.57 | 5.59 | 8.16 |
| | | proto | 43.84 | 37.67 | 20.72 | 28.11 | 26.46 | 62.68 | 22.69 | 22.16 |
| | ViT-L/16 | linear | 12.44 | 16.20 | 6.63 | 8.31 | 15.41 | 41.91 | 5.75 | 7.94 |
| | | proto | **49.97** | **51.73** | **31.38** | **37.80** | **29.97** | **69.50** | **37.74** | **29.96** |
| | ViT-H/16 | linear | 13.25 | 14.82 | 7.29 | 7.93 | 12.99 | 38.71 | 5.60 | 7.84 |
| | | proto | 47.52 | 49.65 | 30.43 | 35.85 | 28.91 | 69.13 | 35.83 | 28.31 |

Table 7: **Probing results on the multi-label classification benchmark with full data (mAP%).** Comparison of linear probing vs. prototypical probing using frozen encoder representations. Models follow the evaluation protocol of BirdSet. **Best** and second best results are highlighted.

## 6.2 Few-Shot Multi-Label Probing

**Settings.** In this section, we evaluate the *few-shot learning* capabilities of Bird-MAE, introducing a few-shot multi-label classification benchmark in BirdSet to test the frozen representations in a low data regime. This setup maintains the standard test sets and domain shifts but restricts the training data to $k \in \{1, 5, 10\}$ event instances per class for each downstream task's training subset. We only report our best-performing Bird-MAE-L model with an average of three repetitions and three randomly sampled subsets per shot. Detailed results are available in Appendix E.2 and the few-shot sampling strategy is described in Appendix A.

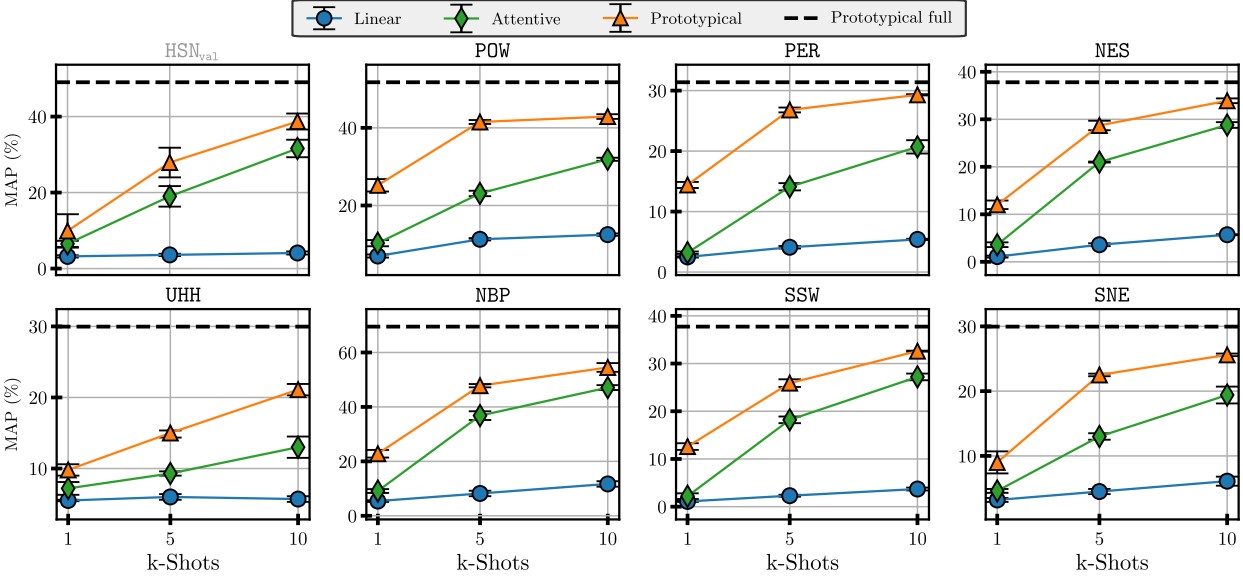

Figure 4: **Few-shot probing results.** We compare linear, attentive and prototypical probes using frozen Bird-MAE-L features at $k \in \{1, 5, 10\}$ shots per class. Results are averaged over three runs on three subsets per shot with the standard deviation. The dashed line marks the upper probing bound on the full dataset.

**How does few-shot prototypical probing compare to other methods?** Figure 4 shows that the advantages of prototypical probing are even more pronounced in low-data regimes. While linear probing yields very low mAP scores across all *k*-shot settings, prototypical probing delivers substantially better performance, even with just one shot per class on the file level. Attentive probing also provides a clear improvement over linear probing but consistently underperforms compared to prototypical probing across all shot counts. These trends are observed uniformly across all eight datasets, underscoring the effectiveness of prototypical probing in leveraging MAE embeddings for few-shot learning.

**Can few-shot prototypical probing rival full dataset probing?** Figure 4 demonstrates that prototypical probing with only 10 shots per class achieves performance close to full-data probing. For instance, on the `PER`, 10-shot prototypical probing reaches approximately 29.31% mAP, approaching the 29.97% mAP of full-data prototypical probing and the 34.64% mAP of full fine-tuning. While attentive probing also shows data efficiency in few-shot settings compared to linear probing, it generally does not reach the same level of proximity to full-dataset performance as prototypical probing. Comparable trends appear across all datasets, illustrating the data efficiency by combining Bird-MAE's features with prototypical probing.

## 7 Discussion and Limitations

Several research directions emerge from our findings, framed by the scope and limitations of this study. In the following, we discuss these limitations and show important directions for future work.

**Upfront cost of domain-specific pretraining.** We advocate for a domain-specific foundation model in complex fields like bioacoustics. This approach requires a notable one-time investment for pretraining, which we frame as a necessary trade-off for achieving SOTA performance in challenging fine-grained domains where general-purpose models currently fall short (Hamer et al., 2023; Turian et al., 2022). This initial cost is balanced by a reusable asset for the research community that enables efficient downstream adaptation (Ghani et al., 2023): Various sub-tasks can be tackled by simply training a lightweight probe, avoiding the need for repeated, costly fine-tuning. For instance, Bird-MAE could be deployed for other bioacoustic tasks like fine-grained call-type classification (Kahl et al., 2021) or population density estimation (Navine et al., 2024). Furthermore, our preliminary results in Table 8 suggest this benefit may even possess cross-species transferability within bioacoustics, as Bird-MAE outperforms the general-purpose model on the `MeerKAT` dataset. Future work could focus on making this domain-specific pretraining more computationally efficient.

| Model | Head | Probing | Fine-tuning |
|---|---|---|---|
| Audio-MAE | `Linear` | 13.19 | 37.3 |
| Audio-MAE | `Proto` | **17.97** | **38.6** |
| BEATs | `Linear` | – | 38.3 |

(a) `AS-20k`

| Model | Arch. | Head | Probing |
|---|---|---|---|
| Audio-MAE | ViT-B | `Linear` | 16.16 |
| Audio-MAE | ViT-B | `Proto` | 28.30 |
| Bird-MAE | ViT-B | `Linear` | 17.38 |
| Bird-MAE | ViT-B | `Proto` | **34.58** |

(b) `MeerKAT`

Table 8: **Generalizability of prototypical layers and Bird-MAE (mAP%).** Performance comparison on two non-bird datasets: general audio (`AS-20k`) and another fine-grained bioacoustic task (`MeerKAT`). We evaluate the impact of using our prototypical head versus a standard linear head for both frozen feature probing and full fine-tuning.

**Recipe and results transferability.** While our pretraining recipe (e.g., extended epochs, mixup) and prototypical probing are holistically adapted for bird bioacoustics, we do not evaluate their transferability to general audio tasks. To investigate the broader applicability of prototypical probing, we present preliminary experiments on two non-bird datasets: the general audio benchmark `AS-20k` and another fine-grained bioacoustic task, the `MeerKAT` mammal vocalization dataset. Details on the `MeerKAT` dataset are provided in Appendix G. Table 8 provides initial evidence of generalizability. On general audio (`AS-20k`), simply replacing the standard linear head with our prototypical pooling layer notably improves the performance of the off-the-shelf Audio-MAE for both frozen feature probing and full fine-tuning. This fine-tuning result surpasses even the more advanced BEATs (Chen et al., 2023). On the related bioacoustic `MeerKAT` task,

we observe two key results: First, prototypical probing again substantially outperforms linear probing for both the general Audio-MAE (+12.1 pp) and our Bird-MAE (+17.2 pp). Second, Bird-MAE outperforms the general Audio-MAE with both probes, supporting findings from Ghani et al. (2023) that bioacoustic pretraining can offer benefits across related fine-grained animal vocalization tasks. These results suggest that our prototype-based methods are not limited to bird sound classification and can unlock greater performance from existing SSL backbones. A comprehensive cross-domain study and extending these methods to other fine-grained settings such as insect call or speaker dialect classification remain important next steps.

**Shortfall of general-purpose MAEs.** While our study demonstrates that a domain-specific and holistically adapted MAE pipeline can excel, there could be other reasons for the performance gap. Our work leaves open questions regarding other factors, such as (1) the model scale or the (2) the SSL objective. Regarding model scale, one might hypothesize that simply using a vastly larger general-purpose model would suffice. However, evidence suggests this approach has limitations. We observe diminishing performance returns when scaling our own adapted model from a ViT-Large to a ViT-Huge backbone. This aligns with findings from benchmarks like HEAR (Turian et al., 2022), where even billion-parameter general audio models have been shown to underperform on specific bioacoustic tasks. Regarding the SSL objective, another hypothesis is that the MAE reconstruction task is inherently less suited for this fine-grained domain than other methods. Evidence suggests the challenge is broader than one specific method. Our study shows that a holistically adapted Bird-MAE notably outperforms other domain-adapted SSL models that use different objectives, including the contrastive SimCLR and the masking-based AVES Table 6. This aligns with findings from BIRB bioacoustic benchmarks (Hamer et al., 2023), where the YamNET model (weak clip-level training) also exhibits performance limitations. Collectively, this suggests that while model scale and SSL objective influence feature quality, a comprehensive, domain-specific adaptation of the entire pipeline appears to be a dominant factor

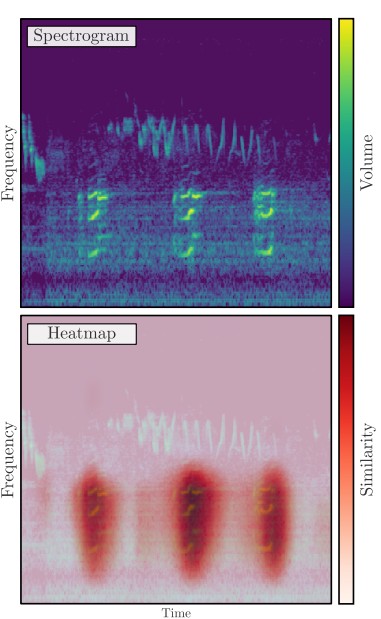

Figure 5: **Activation heatmap** of a prototype superimposed on a spectrogram (Heinrich et al., 2025).

for achieving SOTA performance in this fine-grained domain. Nevertheless, a systematic cross-paradigm comparison on a standardized benchmark like `BirdSet` remains a valuable direction for future work to definitively isolate the impact of different SSL objectives and model sizes.

**Prototype interpretability.** The interpretability of the learned prototypes by heat map visualizations provide a notable value beyond performance metrics. Heinrich et al. conduct a comprehensive analysis demonstrating that prototypes can learn to represent human-interpretable sound patterns, such as distinct call types within a single bird species (as we illustrate exemplary Figure 5). While a detailed interpretability analysis of our specific Bird-MAE representations is beyond the scope of this performance-focused study, it represents an important direction for future research. For instance, the ability to inspect what a model has learned offers opportunities for human-in-the-loop learning, where ecologists could validate or refine prototypes to correct model errors and potentially further enhance performance in data-scarce scenarios (Rauch et al., 2024a). This synergy between strong few-shot performance and interpretability could enable the development of robust and transparent monitoring of of rare species with minimal labels.

## 8 Conclusion

In this work, we addressed the limitations of general-purpose SSL models in audio classification. We demonstrated the efficacy of a holistically adapted, domain-specific masked image modeling pipeline for bird sound classification: We revised the entire training process, including pretraining (e.g., replacing AudioSet with BirdSet), fine-tuning (e.g., adding prototypical pooling), frozen representations (e.g., utilizing prototypical probing), leading to the development of our Bird-MAE model. Bird-MAE achieves novel state-of-the-art performance on the BirdSet multi-label classification benchmark, strongly outperforming the general-purpose

Audio-MAE and prior best-performing supervised models. Our findings highlight that while domain-specific pretraining is crucial, the full benefits of such adaptations become particularly evident when leveraging frozen representations. Specifically, our parameter-efficient prototypical probing substantially narrows the gap to full fine-tuning to 3 pp mAP on average across downstream tasks and boosts it up to 37 pp over linear probing. These results underscore the importance of domain-aware pretrained features and effective probing methods. Furthermore, Bird-MAE with prototypical probing delivers strong few-shot performance, offering an efficient alternative for resource-constrained bioacoustic applications. Our study shows that achieving optimal results in more fine-grained audio tasks such as bioacoustics requires moving beyond generic SSL approaches towards holistic, domain-aware pipeline adaptations.

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

# A  Data Curation

This appendix details our data curation pipeline, a component for both the training of our models and the evaluation of their performance. We apply distinct curation strategies for: (1) the large-scale pretraining dataset, (2) the few-shot learning subsets, and (3) the full datasets for downstream task evaluation. The specifics of each methodology are outlined below.

## A.1  Pretraining Data

We derive our pretraining set from BirdSet's `XCL` collection using the provided file-level species labels, the event detector and the sampling algorithm from BirdSet (Rauch et al., 2024b). Since each recording may contain multiple bird-call events, we first split every recording into its detected events. To mitigate class imbalance, where some species have many events and others few, we enforce three constraints: (a) a maximum number of events per species, (b) a per-recording event cap, and (c) at least one event per recording. We implement these rules via a simple sampling algorithm that iteratively trims over-represented recordings until all class and event limits are met. We set the number of maximum events per species to 500 and a per-recording event cap to 2. This leads to our `XCL-1.7M` pretraining dataset. The uncurated dataset `XCL-3.4M`$_R$ contains all the detected events.

## A.2  Few-shot Data

For our few-shot learning evaluation, we construct k-shot subsets with $k \in \{1, 5, 10\}$ from each BirdSet downstream training split so that every species contributes exactly $k$ audio clips (on file-level). To assess sampling variability, we generate three independent subsets per $k$ using different random seeds. Because BirdSet's labels are weakly labeled, we prioritize recordings under 5 seconds to reduce label noise (i.e., increasing the chance that the annotated species on file-level is actually present in each extracted event). In the following, we describe the $k$-shot subset creation pipeline:

1.  **Initial filtering:** We first go through all recordings in the original training split for a given BirdSet task. The preferred recordings are up to 5 seconds, aligning with the 5-second input file length of the model to mitigate label noise. However, since there are not always even 5-second recordings for each species, we also include 20 seconds samples, but only if they contained just one primary bird species (no secondary species listed).

2.  **Sample extraction from recording:** From each selected recording (which can contain multiple vocalization events), we extract individual 5-second audio samples centered around these events based on the given events from the BirdSet metadata. To avoid over-representing any single long recording, if a recording yields multiple 5-second samples, one is randomly chosen as a *primary sample* for that recording, and the others are considered *leftover samples*.

3.  **Selecting k samples per class:** We first try to pick $k$ samples from the *primary sample* associated with that species (if available). If there are not enough *primary samples* (less than k), we then try to fill the remaining spots using the *leftover samples* from that species. If, a class still has fewer than $k$ samples, we do not fill up further from recordings that failed the initial filtering. This means some classes in the few-shot set might have fewer than $k$ samples if not enough suitable recordings are available. If a species has more than $k$ suitable samples, we randomly selected $k$ samples from the available pool for that species.

4.  **Dataset construction:** The selected $k$ samples per class form the new train split for that specific $k$-shot, seeded dataset. The original BirdSet test split (5-second segments) is kept as is for evaluation.

## A.3  Full Downstream Data

The BirdSet dataset provides file-level (weak) labels for recordings that may contain multiple vocalization events, necessitating downstream processing to generate task-specific datasets (Rauch et al., 2024b). Our

preliminary experiments on small focal validation sets on all downstream tasks and models indicated that a uniform sampling strategy across all BirdSet's tasks is suboptimal. Consequently, we adopted tailored approaches: For the datasets `HSN`, `UHH`, and `NBP` that have the lowest class counts across tasks, we utilized BirdSet's inherent sampling strategy by adding a species cap of 500 and extracting a maximum of 5 events per recording, following Rauch et al. (2024b). More details are available in the BirdSet implementations. Conversely, the datasets `SNE`, `POW`, `NES`, `PER`, and `SSW` are characterized by larger data volumes and higher class counts. We found it more advantageous to employ our few-shot sampling approach with $k = 64$ to further reduce label noise and class imbalance. This proved beneficial for experiments involving both fine-tuning and frozen representations across all models. The final number of samples curated for each dataset in our study is detailed in the main text in Table 2.

## B  Metrics

This appendix provides a detailed description of the evaluation metrics employed to assess model performance throughout this study. We evaluate model performance in the main paper with the mean average precision (MAP) as the metric in multi-label classification (Huang et al., 2022; Chen et al., 2023; 2024a; Rauch et al., 2024b). In the additional results in Appendix E, we also report the area under the receiver operating characteristic curve (AUROC), and top-1 accuracy (T1-Acc), following the multi-label benchmark from BirdSet.

- **MAP**, also referred to as class-wise MAP (cmAP) (Rauch et al., 2024b), first calculates the average precision (AP) for each class $c$ independently and then computes the macro-average of these AP scores across all $C$ classes:

$$\text{MAP} = \frac{1}{C} \sum_{c=1}^{C} \text{AP}(c). \tag{2}$$

  MAP reflects the model's ability to rank positive instances higher than negative ones for each class across all decision thresholds, providing a comprehensive assessment of retrieval performance. By averaging class-wise AP scores, MAP gives equal weight to each class, regardless of its prevalence in the dataset. While robust, it can be sensitive to classes with very few positive instances (van Merriënboer et al., 2024).

- **AUROC** quantifies the model's ability to discriminate between positive and negative instances across all possible classification thresholds. It is equivalent to the probability that a randomly chosen positive instance is ranked higher by the model than a randomly chosen negative instance (Heinrich et al., 2025). For a multi-label setting, it is often computed as the average AUROC across all classes:

$$\text{AUROC} = \frac{1}{C} \sum_{c=1}^{C} \left( \frac{1}{|Y_{+,c}| \cdot |Y_{-,c}|} \sum_{n \in Y_{+,c}} \sum_{m \in Y_{-,c}} \mathbb{I}\{\hat{y}_{n,c} > \hat{y}_{m,c}\} \right),$$

  where $Y_{+,c}$ and $Y_{-,c}$ are the sets of indices for positive and negative instances for class $c$ respectively, $\hat{y}_{n,c}$ is the predicted score for instance $n$ and class $c$, and $\mathbb{I}\{\cdot\}$ is the indicator function. AUROC is threshold-independent and provides a balanced view of performance, where a random classifier yields an AUROC of 0.5 (van Merriënboer et al., 2024).

- **T1-Acc** assesses whether the class assigned the highest predicted confidence score by the model is among the set of true labels for a given instance (Rauch et al., 2024b):

$$\text{T1-Acc} = \frac{1}{N} \sum_{n=1}^{N} \mathbb{I}\{\hat{y}_n^{(\text{top})} \in Y_{n,:}\},$$

  where $N$ is the total number of instances, $\hat{y}_n^{(\text{top})}$ is the class with the highest predicted score for instance $n$, $Y_{n,:}$ is the set of true labels for instance $n$, and $\mathbb{I}\{\cdot\}$ is the indicator function. While not a canonical multi-label metric, T1-Acc offers an intuitive measure of whether the model's most confident prediction is correct, which is relevant in practical applications where identifying at least one present species is a primary goal.

## C   Implementation and Infrastructure

To facilitate reproduction, experiments were run under the following conditions: Models were trained and evaluated on a compute cluster using NVIDIA L40s and A100 GPUs. CPU types included Intel Xeon Gold 6252 and AMD EPYC 7662, with nodes having approximately 600 GB RAM. The software environment consisted of Ubuntu OS, Python 3.9, PyTorch (Paszke et al., 2019), and PyTorch Lightning (Falcon & The PyTorch Lightning team, 2019). Small-scale testing was performed on a workstation using an NVIDIA RTX 4090 GPU and an AMD Ryzen 9 7950X CPU.

## D   Model Training

This appendix outlines the specific training configurations and hyperparameters employed for our experiments, including both the ablation studies and the final benchmark evaluations.

### D.1   Augmentations

Our data augmentation pipeline, applied during to all experiments, including few-shot multi-label classification and probing variants, is adapted from the strategies outlined in BirdSet (Rauch et al., 2024b). We empirically tuned the application probability for each selected augmentation based on preliminary experiments on the validation data `HSN`. The spectrogram time-frequency masking is similar to SpecAugment (Park et al., 2019). However, we omit the time-warping component of SpecAugment, which did not improve validation performance in our setting. A key component of our pipeline is waveform-level mixup, implemented using TorchAudiomentations (Jordal et al., 2024), which demonstrated superior performance compared to spectrogram-based or standard linear mixup. For augmentations requiring external audio, such as background noise addition and no-call mixing, we utilized environmental recordings sourced from BirdSet's `VOX` dataset. Additional waveform and spectrogram-level augmentations, along with their specific parameters, are detailed in Table 9.

| Augmentation | Probability | Parameters |
|---|---|---|
| *Waveform-level augmentations* | | |
| `cyclic rolling start` | 1.0 | - |
| `multi-label mixup` | 0.9 | min-snr=2.0, max-snr=30.0, mix-target=union, max-samples=3 |
| `background noise` | 0.5 | min-snr=3.0, max-snr=30.0 |
| `colored noise` | 0.2 | min-snr=3.0, max-snr=30.0, min-f-decay=-2, max-f-decay=2 |
| `gain adjustment` | 0.2 | min-gain=-18, max-gain=6 |
| `no-call mixing` | 0.075 | - |
| *Spectrogram-level augmentations* | | |
| `frequency masking` | 0.3 | freq-mask-param=50, iid-masks=True |
| `time masking` | 0.3 | time-mask-param=100, iid-masks=True |

Table 9: **Data augmentation techniques and parameters** applied during all experiments in the paper. This includes fine-tuning and probing on the complete dataset as well as few-shot probing across all techniques.

### D.2   Hyperparameters

This section details the hyperparameters used for fine-tuning our models in both the ablation studies and the main `BirdSet` benchmark experiments. These parameters, including learning rates, batch sizes, and optimizers, were empirically validated on the `HSN` dataset. After validation, the hyperparameters were fixed across all downstream tasks. All models utilize the asymmetric loss for multi-label classification (Ridnik et al., 2021). The same core settings were also applied to the few-shot learning benchmark, with minor adjustments primarily to the learning rate and number of training epochs to suit the reduced data regime.

For each model, whether we fine tune it on the full data or probe frozen representations, we validate two hyperparameters: the learning rate and the weight decay. With prototypical pooling or probing we additionally explore the number of prototypes $J$ (see Table 13) and the learning rate of the prototype vectors. Every model and setting combination is trained for 30 epochs in the full data regime and 50 epochs in the few shot regime, using random search over these discrete grids:

- Learning rate: $\{1 \times 10^{-5}, 1 \times 10^{-4}, 2 \times 10^{-4}, 3 \times 10^{-4}, 4 \times 10^{-4}, 5 \times 10^{-4}, 1 \times 10^{-3}, 5 \times 10^{-3}\}$

- Weight decay: $\{1 \times 10^{-4}, 2 \times 10^{-4}, 3 \times 10^{-4}, 4 \times 10^{-4}, 5 \times 10^{-4}\}$

- Number of prototypes $J$: $\{5, 10, 15, 20, 25, 30\}$

- Prototype learning rate: $\{2 \times 10^{-2}, 4 \times 10^{-2}, 5 \times 10^{-2}\}$

| Parameter | BirdAVES (Hagiwara, 2023) | SimCLR (Moummad et al., 2024) | Audio-MAE (Huang et al., 2022) | Bird-MAE |
|---|---|---|---|---|
| *Model Training (Fine-Tuning / Probing on full data)* | | | | |
| Input Type | Waveform | Spectrogram | Spectrogram (fbank) | Spectrogram (fbank) |
| Batch size | 64 | 128 | 128 | 128,128,64 |
| # Epochs | 30 | 30 | 30 | 30 |
| Gradient clip | 0.5 | 0.1 | 2 | 2 |
| Precision | 16-mixed | 16-mixed | 16-mixed | 16-mixed |
| Learning rate | 1e-5 | 4e-4 | 3e-4 | 3e-4 |
| Optimizer | AdamW | AdamW | AdamW | AdamW |
| Loss | Asymmetric | Asymmetric | Asymmetric | Asymmetric |
| Weight decay | 1e-4 | 3e-4 | 3e-4 | 3e-4 |
| Layer decay | - | - | 0.75 | 0.75 |
| Pooling | mean | mean | prototypical | prototypical |
| Scheduler | Cos-Annealing | Cos-Annealing | Cos-Annealing | Cos-Annealing |
| *Model Training (Few-shot probing)* | | | | |
| Learning Rate | - | - | - | 4e-4 |
| Epochs | - | - | - | 50 |
| *Processing* | | | | |
| # fft | - | 1024 | $\sim$800 | $\sim$800 |
| # Mels | - | 128 | 128 | 128 |
| Sampling Rate | 16 kHz | 16 kHz | 16 kHz | 32 kHz |
| Norm Mean | - | 0.5347 | -4.2 | -7.2 |
| Norm Std | - | 0.0772 | 4.57 | 4.43 |
| Fmin-Fmax | - | 50-8000 Hz | full band | full band |
| Window Type | - | Hanning | Hanning | Hanning |
| Frame Shift | - | 320 samples | 10 ms | 10 ms |
| Log Scale | - | AmplitudeToDB | log FBANK | log FBANK |
| *Pretraining* | | | | |
| Type | SSL | SSL | SSL | SSL |
| Architecture | HuBERT-L | CvT-13 | ViT-B/16 | ViT-{B,L,H}/16 |
| # Params [M] | 317 | 20 | 86 | 86,307,632 |
| Dataset | Xeno-Canto* | Xeno-Canto* | AS-2M | XCL-1.7M |
| *Prototypical Pooling / Probing* | | | | |
| # Prototypes | 20 | 20 | 20 | 20 |
| Protoype LR | 4e-2 | 4e-2 | 4e-2 | 4e-2 |
| Focal Similarity | True | True | True | True |
| Orthogonality Loss | True | True | True | True |
| *Attentive Pooling / Probing* | | | | |
| # Attention heads | - | - | 12 | 12 |

Table 10: **Training hyperparameters for models evaluated in this paper.** These settings cover evaluations using frozen representations, full fine-tuning (ablation studies), and both techniques for multi-label classification on the benchmark results. For the multi-label few-shot classification benchmark, we largely retained the same hyperparameter settings, with specific adjustments primarily to the learning rate and number of training epochs.

# E  Additional Benchmark Results

This appendix presents complementary results to the main multi-label and few-shot classification benchmarks discussed in the main text. These additional evaluations provide further insights into model performance with BirdSet's metric suite.

## E.1  Multi-Label Classification

| | Model | Fine-Tuning | | | Prototypical Probing | | | Linear Probing | | |
|---|---|---|---|---|---|---|---|---|---|---|
| | | AUROC | MAP | T1-Acc | AUROC | MAP | T1-Acc | AUROC | MAP | T1-Acc |
| HSN | BirdAVES | 0.86 ± 0.03 | 0.43 ± 0.01 | 0.55 ± 0.03 | 0.79 ± 0.00 | 0.33 ± 0.00 | 0.25 ± 0.01 | **0.81 ± 0.00** | 0.15 ± 0.00 | **0.13 ± 0.00** |
| | SimCLR | 0.83 ± 0.00 | 0.39 ± 0.00 | 0.54 ± 0.01 | 0.73 ± 0.01 | 0.18 ± 0.00 | 0.18 ± 0.01 | 0.70 ± 0.00 | **0.17 ± 0.00** | 0.12 ± 0.00 |
| | Audio-MAE | 0.83 ± 0.00 | 0.19 ± 0.00 | 0.10 ± 0.00 | 0.19 ± 0.00 | 0.10 ± 0.00 | 0.10 ± 0.00 | 0.68 ± 0.00 | 0.09 ± 0.00 | 0.06 ± 0.00 |
| | Bird-MAE-B | 0.86 ± 0.00 | 0.52 ± 0.01 | 0.58 ± 0.01 | 0.89 ± 0.01 | 0.44 ± 0.05 | 0.37 ± 0.16 | 0.77 ± 0.00 | 0.13 ± 0.00 | 0.07 ± 0.01 |
| | Bird-MAE-L | **0.90 ± 0.00** | **0.55 ± 0.00** | 0.64 ± 0.01 | **0.90 ± 0.00** | **0.49 ± 0.01** | 0.38 ± 0.01 | 0.72 ± 0.00 | 0.12 ± 0.00 | 0.07 ± 0.00 |
| | Bird-MAE-H | **0.91 ± 0.00** | **0.55 ± 0.00** | **0.65 ± 0.00** | **0.91 ± 0.00** | 0.48 ± 0.01 | **0.46 ± 0.08** | 0.76 ± 0.00 | 0.13 ± 0.00 | 0.10 ± 0.00 |
| POW | BirdAVES | 0.60 ± 0.01 | 0.11 ± 0.02 | 0.10 ± 0.03 | 0.66 ± 0.00 | 0.20 ± 0.01 | 0.28 ± 0.03 | 0.68 ± 0.01 | 0.15 ± 0.00 | 0.31 ± 0.04 |
| | SimCLR | 0.81 ± 0.00 | 0.35 ± 0.01 | 0.67 ± 0.04 | 0.64 ± 0.01 | 0.17 ± 0.00 | 0.27 ± 0.05 | **0.71 ± 0.01** | **0.18 ± 0.00** | **0.39 ± 0.01** |
| | Audio-MAE | 0.82 ± 0.01 | 0.39 ± 0.01 | 0.75 ± 0.00 | 0.74 ± 0.00 | 0.20 ± 0.00 | 0.26 ± 0.02 | 0.60 ± 0.00 | 0.10 ± 0.00 | 0.09 ± 0.01 |
| | Bird-MAE-B | 0.86 ± 0.00 | 0.45 ± 0.01 | 0.83 ± 0.00 | 0.85 ± 0.01 | 0.38 ± 0.01 | 0.74 ± 0.02 | 0.67 ± 0.00 | 0.13 ± 0.00 | 0.23 ± 0.06 |
| | Bird-MAE-L | **0.90 ± 0.00** | **0.55 ± 0.01** | **0.91 ± 0.00** | **0.90 ± 0.00** | **0.52 ± 0.00** | **0.89 ± 0.01** | 0.67 ± 0.01 | 0.14 ± 0.00 | 0.22 ± 0.06 |
| | Bird-MAE-H | 0.89 ± 0.00 | 0.54 ± 0.01 | 0.89 ± 0.00 | 0.89 ± 0.01 | 0.50 ± 0.01 | 0.86 ± 0.01 | 0.67 ± 0.00 | 0.15 ± 0.00 | 0.32 ± 0.04 |
| NES | BirdAVES | 0.75 ± 0.03 | 0.06 ± 0.00 | 0.13 ± 0.02 | 0.65 ± 0.01 | 0.12 ± 0.01 | 0.23 ± 0.03 | 0.70 ± 0.00 | 0.08 ± 0.00 | 0.12 ± 0.02 |
| | SimCLR | 0.85 ± 0.01 | 0.24 ± 0.01 | 0.39 ± 0.01 | 0.62 ± 0.01 | 0.08 ± 0.01 | 0.20 ± 0.02 | 0.75 ± 0.00 | 0.11 ± 0.00 | 0.11 ± 0.01 |
| | Audio-MAE | 0.87 ± 0.00 | 0.30 ± 0.00 | 0.42 ± 0.01 | 0.87 ± 0.00 | 0.16 ± 0.00 | 0.25 ± 0.01 | 0.70 ± 0.00 | 0.04 ± 0.00 | 0.07 ± 0.01 |
| | Bird-MAE-B | 0.90 ± 0.00 | 0.37 ± 0.00 | 0.47 ± 0.00 | 0.92 ± 0.00 | 0.28 ± 0.00 | 0.41 ± 0.01 | **0.78 ± 0.00** | 0.08 ± 0.00 | 0.11 ± 0.01 |
| | Bird-MAE-L | **0.91 ± 0.00** | **0.41 ± 0.01** | **0.52 ± 0.00** | **0.93 ± 0.00** | **0.38 ± 0.00** | **0.47 ± 0.00** | 0.75 ± 0.00 | 0.06 ± 0.00 | **0.22 ± 0.01** |
| | Bird-MAE-H | 0.89 ± 0.00 | 0.39 ± 0.00 | 0.51 ± 0.01 | 0.92 ± 0.00 | 0.36 ± 0.01 | **0.47 ± 0.01** | 0.75 ± 0.00 | **0.11 ± 0.00** | **0.22 ± 0.01** |
| SNE | BirdAVES | 0.65 ± 0.03 | 0.07 ± 0.00 | 0.04 ± 0.04 | 0.59 ± 0.00 | 0.10 ± 0.01 | 0.27 ± 0.01 | **0.67 ± 0.01** | 0.08 ± 0.00 | **0.20 ± 0.01** |
| | SimCLR | 0.75 ± 0.01 | 0.16 ± 0.00 | 0.40 ± 0.03 | 0.58 ± 0.00 | 0.09 ± 0.01 | 0.12 ± 0.04 | 0.61 ± 0.01 | **0.09 ± 0.00** | 0.18 ± 0.03 |
| | Audio-MAE | 0.81 ± 0.01 | 0.22 ± 0.00 | 0.42 ± 0.01 | 0.75 ± 0.00 | 0.12 ± 0.00 | 0.13 ± 0.04 | 0.62 ± 0.01 | 0.06 ± 0.00 | 0.01 ± 0.01 |
| | Bird-MAE-B | 0.83 ± 0.01 | 0.28 ± 0.01 | 0.52 ± 0.01 | 0.84 ± 0.00 | 0.22 ± 0.00 | 0.35 ± 0.02 | **0.67 ± 0.01** | 0.08 ± 0.00 | 0.04 ± 0.01 |
| | Bird-MAE-L | **0.88 ± 0.00** | **0.34 ± 0.00** | **0.61 ± 0.01** | **0.86 ± 0.00** | **0.30 ± 0.00** | **0.54 ± 0.01** | 0.64 ± 0.01 | 0.08 ± 0.00 | **0.20 ± 0.01** |
| | Bird-MAE-H | 0.85 ± 0.01 | 0.32 ± 0.00 | **0.62 ± 0.01** | 0.85 ± 0.01 | 0.28 ± 0.01 | 0.48 ± 0.02 | 0.61 ± 0.01 | **0.09 ± 0.00** | 0.18 ± 0.03 |
| SSW | BirdAVES | 0.76 ± 0.02 | 0.04 ± 0.00 | 0.10 ± 0.02 | 0.62 ± 0.00 | 0.08 ± 0.03 | 0.20 ± 0.03 | 0.73 ± 0.00 | 0.06 ± 0.00 | 0.16 ± 0.01 |
| | SimCLR | 0.83 ± 0.02 | 0.19 ± 0.02 | 0.44 ± 0.01 | 0.60 ± 0.01 | 0.05 ± 0.00 | 0.16 ± 0.02 | 0.74 ± 0.00 | 0.07 ± 0.00 | 0.18 ± 0.01 |
| | Audio-MAE | 0.89 ± 0.00 | 0.27 ± 0.00 | 0.52 ± 0.01 | 0.86 ± 0.00 | 0.09 ± 0.00 | 0.25 ± 0.02 | 0.69 ± 0.01 | 0.03 ± 0.00 | 0.10 ± 0.01 |
| | Bird-MAE-B | 0.88 ± 0.00 | 0.33 ± 0.01 | 0.62 ± 0.00 | **0.94 ± 0.00** | 0.23 ± 0.00 | 0.49 ± 0.00 | **0.77 ± 0.00** | 0.05 ± 0.00 | 0.14 ± 0.00 |
| | Bird-MAE-L | **0.93 ± 0.00** | **0.41 ± 0.00** | **0.70 ± 0.00** | **0.94 ± 0.00** | **0.38 ± 0.00** | **0.62 ± 0.00** | **0.77 ± 0.00** | 0.05 ± 0.00 | 0.14 ± 0.00 |
| | Bird-MAE-H | 0.91 ± 0.00 | 0.41 ± 0.00 | 0.68 ± 0.00 | **0.94 ± 0.00** | 0.36 ± 0.02 | 0.60 ± 0.02 | 0.74 ± 0.00 | **0.07 ± 0.00** | **0.18 ± 0.01** |
| PER | BirdAVES | 0.58 ± 0.02 | 0.05 ± 0.01 | 0.11 ± 0.02 | 0.54 ± 0.00 | 0.05 ± 0.00 | 0.14 ± 0.00 | 0.60 ± 0.00 | 0.05 ± 0.00 | 0.13 ± 0.01 |
| | SimCLR | 0.73 ± 0.00 | 0.17 ± 0.01 | 0.43 ± 0.01 | 0.52 ± 0.00 | 0.03 ± 0.00 | 0.06 ± 0.02 | 0.63 ± 0.01 | 0.06 ± 0.00 | 0.13 ± 0.01 |
| | Audio-MAE | 0.77 ± 0.00 | 0.21 ± 0.00 | 0.49 ± 0.00 | 0.71 ± 0.00 | 0.09 ± 0.00 | 0.17 ± 0.00 | 0.60 ± 0.00 | 0.04 ± 0.00 | 0.09 ± 0.00 |
| | Bird-MAE-B | 0.77 ± 0.00 | 0.28 ± 0.00 | 0.54 ± 0.00 | 0.79 ± 0.00 | 0.21 ± 0.00 | 0.49 ± 0.01 | 0.63 ± 0.00 | 0.05 ± 0.00 | 0.13 ± 0.00 |
| | Bird-MAE-L | **0.82 ± 0.00** | **0.35 ± 0.00** | **0.60 ± 0.01** | **0.82 ± 0.00** | **0.31 ± 0.00** | **0.59 ± 0.00** | 0.63 ± 0.01 | 0.05 ± 0.00 | 0.13 ± 0.00 |
| | Bird-MAE-H | 0.81 ± 0.00 | 0.33 ± 0.00 | 0.59 ± 0.00 | **0.82 ± 0.00** | 0.30 ± 0.01 | 0.58 ± 0.01 | **0.66 ± 0.01** | **0.07 ± 0.00** | **0.18 ± 0.02** |
| UHH | BirdAVES | **0.82 ± 0.01** | 0.22 ± 0.01 | **0.47 ± 0.02** | 0.71 ± 0.01 | 0.15 ± 0.01 | 0.29 ± 0.00 | 0.72 ± 0.00 | 0.12 ± 0.00 | 0.28 ± 0.00 |
| | SimCLR | 0.76 ± 0.00 | 0.21 ± 0.00 | 0.37 ± 0.01 | 0.57 ± 0.07 | 0.07 ± 0.03 | 0.19 ± 0.08 | **0.76 ± 0.00** | **0.15 ± 0.00** | **0.29 ± 0.00** |
| | Audio-MAE | 0.82 ± 0.00 | 0.26 ± 0.00 | 0.38 ± 0.00 | 0.74 ± 0.00 | 0.17 ± 0.00 | 0.26 ± 0.00 | 0.67 ± 0.00 | 0.11 ± 0.00 | 0.29 ± 0.00 |
| | Bird-MAE-B | 0.81 ± 0.01 | 0.28 ± 0.01 | 0.41 ± 0.04 | **0.83 ± 0.01** | 0.26 ± 0.03 | 0.32 ± 0.04 | **0.76 ± 0.00** | **0.15 ± 0.00** | **0.29 ± 0.00** |
| | Bird-MAE-L | **0.82 ± 0.01** | **0.30 ± 0.00** | 0.42 ± 0.00 | **0.83 ± 0.00** | **0.30 ± 0.00** | **0.36 ± 0.00** | **0.76 ± 0.00** | **0.15 ± 0.00** | 0.28 ± 0.00 |
| | Bird-MAE-H | 0.80 ± 0.01 | **0.30 ± 0.00** | 0.41 ± 0.02 | 0.82 ± 0.01 | 0.29 ± 0.00 | 0.35 ± 0.02 | 0.71 ± 0.05 | 0.13 ± 0.03 | 0.19 ± 0.07 |
| NBP | BirdAVES | 0.93 ± 0.00 | 0.70 ± 0.00 | **0.78 ± 0.01** | 0.79 ± 0.02 | 0.40 ± 0.01 | 0.45 ± 0.01 | 0.79 ± 0.00 | 0.34 ± 0.00 | 0.37 ± 0.01 |
| | SimCLR | 0.92 ± 0.00 | 0.66 ± 0.00 | 0.73 ± 0.01 | 0.73 ± 0.01 | 0.27 ± 0.01 | 0.30 ± 0.01 | 0.73 ± 0.00 | 0.34 ± 0.00 | 0.37 ± 0.01 |
| | Audio-MAE | 0.93 ± 0.00 | 0.67 ± 0.01 | 0.68 ± 0.01 | 0.81 ± 0.00 | 0.35 ± 0.00 | 0.37 ± 0.00 | 0.75 ± 0.00 | 0.25 ± 0.00 | 0.21 ± 0.00 |
| | Bird-MAE-B | 0.91 ± 0.00 | 0.63 ± 0.01 | 0.66 ± 0.02 | **0.92 ± 0.03** | 0.63 ± 0.07 | 0.65 ± 0.06 | 0.79 ± 0.00 | 0.34 ± 0.00 | **0.45 ± 0.01** |
| | Bird-MAE-L | **0.94 ± 0.00** | **0.72 ± 0.00** | 0.72 ± 0.01 | **0.92 ± 0.00** | **0.69 ± 0.00** | **0.69 ± 0.01** | **0.92 ± 0.00** | **0.42 ± 0.00** | 0.42 ± 0.00 |
| | Bird-MAE-H | **0.94 ± 0.00** | 0.69 ± 0.00 | 0.70 ± 0.01 | **0.92 ± 0.01** | 0.69 ± 0.02 | 0.69 ± 0.02 | 0.80 ± 0.00 | 0.25 ± 0.00 | 0.21 ± 0.00 |

Table 11: **Fine-tuning, prototypical probing and linear probing** results on BirdSet's *multi-label classification benchmark* (MAP, AUROC, T1-Acc.). Comparison of SSL models with *full training data*, following the evaluation protocol of BirdSet. Best results are highlighted. This complements Table 6 from the main text.

## E.2 Few-Shot Multi-Label Classification

| | | 1-shot | | | 5-shot | | | 10-shot | | |
|---|---|---|---|---|---|---|---|---|---|---|
| | | AUROC | MAP | T1-Acc | AUROC | MAP | T1-Acc | AUROC | MAP | T1-Acc |
| POW | Proto | **72.05 ± 0.86** | **25.22 ± 1.55** | **45.15 ± 7.33** | **83.75 ± 0.36** | **41.46 ± 0.53** | **74.37 ± 1.11** | **85.25 ± 0.36** | **42.95 ± 0.60** | **87.61 ± 0.53** |
| | Linear | 52.26 ± 1.73 | 7.05 ± 0.36 | 9.93 ± 9.48 | 57.01 ± 0.26 | 11.25 ± 0.32 | 7.26 ± 3.89 | 62.05 ± 0.49 | 12.53 ± 0.28 | 26.15 ± 10.65 |
| | Attentive | 57.45 ± 1.41 | 10.28 ± 0.83 | 6.14 ± 4.26 | 70.35 ± 1.20 | 23.11 ± 0.70 | 54.30 ± 8.62 | 75.89 ± 0.64 | 31.88 ± 0.40 | 75.88 ± 2.01 |
| HSN | Proto | **60.55 ± 7.23** | **9.91 ± 4.44** | 2.03 ± 1.94 | **82.54 ± 1.85** | **27.88 ± 3.91** | **15.95 ± 6.26** | **86.35 ± 0.56** | **38.65 ± 2.08** | 21.58 ± 2.52 |
| | Linear | 52.46 ± 2.18 | 3.21 ± 0.28 | **5.39 ± 6.84** | 54.62 ± 1.17 | 3.57 ± 0.29 | 0.67 ± 0.23 | 55.10 ± 1.87 | 4.06 ± 0.44 | 1.15 ± 0.61 |
| | Attentive | 60.03 ± 2.82 | 6.48 ± 0.80 | 3.20 ± 3.22 | 71.26 ± 1.75 | 19.01 ± 2.75 | 11.97 ± 3.01 | 80.26 ± 0.94 | 31.58 ± 2.35 | **22.01 ± 2.67** |
| PER | Proto | **70.50 ± 0.69** | **14.36 ± 0.50** | **18.81 ± 7.67** | **80.00 ± 0.37** | **26.83 ± 0.40** | **58.94 ± 2.42** | **81.22 ± 0.14** | **29.33 ± 0.11** | **62.10 ± 1.26** |
| | Linear | 52.96 ± 1.31 | 2.49 ± 0.05 | 4.89 ± 5.36 | 60.09 ± 1.03 | 4.09 ± 0.21 | 4.94 ± 2.14 | 63.01 ± 1.07 | 5.41 ± 0.12 | 7.02 ± 0.70 |
| | Attentive | 55.89 ± 1.03 | 3.17 ± 0.24 | 2.15 ± 1.12 | 68.45 ± 1.83 | 14.08 ± 0.56 | 39.55 ± 2.15 | 74.22 ± 1.06 | 20.69 ± 1.08 | 50.83 ± 2.29 |
| NES | Proto | **78.78 ± 1.27** | **12.03 ± 0.93** | **13.70 ± 1.56** | **88.97 ± 0.80** | **28.65 ± 1.02** | **41.11 ± 2.10** | **91.44 ± 0.31** | **33.86 ± 0.51** | **46.91 ± 0.42** |
| | Linear | 55.95 ± 1.73 | 1.06 ± 0.17 | 0.29 ± 0.17 | 64.77 ± 0.97 | 3.65 ± 0.31 | 2.64 ± 1.17 | 68.96 ± 0.73 | 5.71 ± 0.11 | 7.82 ± 0.94 |
| | Attentive | 62.20 ± 1.52 | 3.58 ± 0.46 | 2.89 ± 1.57 | 83.38 ± 0.90 | 20.98 ± 0.13 | 32.76 ± 4.37 | 87.76 ± 0.76 | 28.78 ± 0.64 | 42.53 ± 1.44 |
| UHH | Proto | **67.89 ± 1.30** | **9.82 ± 0.82** | **10.59 ± 2.02** | **73.73 ± 0.70** | **14.96 ± 0.52** | **35.36 ± 0.44** | **75.24 ± 0.63** | **21.06 ± 0.80** | **48.78 ± 1.97** |
| | Linear | 57.27 ± 2.36 | 5.54 ± 0.09 | 8.92 ± 12.29 | 58.21 ± 1.85 | 6.04 ± 0.36 | 22.25 ± 5.36 | 60.18 ± 2.22 | 5.74 ± 0.36 | 17.72 ± 7.57 |
| | Attentive | 60.62 ± 0.86 | 7.23 ± 0.85 | 8.74 ± 2.80 | 65.01 ± 1.01 | 9.26 ± 0.33 | 17.74 ± 5.48 | 66.57 ± 1.58 | 13.00 ± 1.53 | 45.12 ± 2.22 |
| NBP | Proto | **69.10 ± 0.94** | **22.81 ± 1.35** | **17.89 ± 3.92** | **84.51 ± 0.71** | **47.81 ± 0.60** | **49.13 ± 2.81** | **86.37 ± 1.11** | **54.53 ± 1.60** | **55.88 ± 1.26** |
| | Linear | 53.09 ± 0.35 | 5.44 ± 0.26 | 4.08 ± 0.40 | 58.00 ± 0.51 | 8.18 ± 1.02 | 4.33 ± 0.76 | 61.96 ± 0.24 | 11.69 ± 1.05 | 7.73 ± 0.89 |
| | Attentive | 58.05 ± 1.02 | 9.08 ± 0.67 | 5.63 ± 0.95 | 77.83 ± 0.74 | 36.80 ± 1.63 | 41.25 ± 2.22 | 82.58 ± 0.76 | 47.06 ± 0.93 | 55.35 ± 1.79 |
| SSW | Proto | **81.33 ± 2.53** | **12.61 ± 0.68** | **26.87 ± 3.96** | **90.08 ± 0.28** | **25.87 ± 0.80** | **45.11 ± 0.98** | **92.54 ± 0.17** | **32.64 ± 0.13** | **50.01 ± 0.54** |
| | Linear | 61.94 ± 2.05 | 1.15 ± 0.09 | 0.38 ± 0.32 | 67.25 ± 1.65 | 2.35 ± 0.23 | 6.07 ± 3.48 | 68.00 ± 0.79 | 3.68 ± 0.30 | 12.16 ± 0.29 |
| | Attentive | 55.86 ± 0.95 | 2.25 ± 0.64 | 3.42 ± 1.53 | 79.93 ± 0.87 | 18.16 ± 0.73 | 38.35 ± 0.54 | 85.81 ± 0.48 | 27.20 ± 0.65 | 49.50 ± 1.27 |
| SNE | Proto | **68.86 ± 0.90** | **9.05 ± 1.66** | **7.23 ± 3.56** | **81.03 ± 0.25** | **22.53 ± 0.20** | **38.15 ± 0.40** | **83.01 ± 0.74** | **25.60 ± 0.19** | **47.15 ± 2.48** |
| | Linear | 54.18 ± 1.86 | 3.19 ± 0.27 | 0.57 ± 0.51 | 56.28 ± 1.06 | 4.53 ± 0.44 | 2.38 ± 1.27 | 58.56 ± 1.16 | 6.09 ± 0.68 | 4.40 ± 1.58 |
| | Attentive | 53.39 ± 2.00 | 4.56 ± 0.33 | 2.80 ± 1.61 | 68.66 ± 1.80 | 13.01 ± 0.49 | 27.63 ± 2.57 | 75.99 ± 0.97 | 19.43 ± 1.31 | 37.30 ± 4.32 |

Table 12: **Frozen representation results for prototypical, linear and attentive probing** on our *few-shot multi-label classification benchmark* (MAP, AUROC, T1-Acc.). Comparison of our best-performing Bird-MAE-L model with *few-shot training data*, following the evaluation protocol of (Rauch et al., 2024b). **Best** results are highlighted. This complements Table 7 from the main text.

# F Additional Ablations

This appendix contains supplementary ablation studies. As in the main text, all ablations are performed on the HSN validation set with the best-performing model modifications and parameters.

| $J$ | Probing | Fine-tuning |
|---|---|---|
| 5 | 39.16 | 52.22 |
| 10 | 43.78 | 54.07 |
| 15 | 47.47 | 54.16 |
| 20 | 49.92 | 54.91 |
| 25 | 49.92 | 54.91 |
| 30 | 49.97 | 54.71 |

Table 13: **Ablation on number of prototypes** ($J$) on HSN with the Bird-MAE-L model (MAP%).

# G Generalizability Study on MeerKAT

To investigate the generalizability of our findings beyond avian bioacoustics, we conduct a preliminary study on the MeerKAT dataset (Schäfer-Zimmermann et al., 2024), which contains multi-label meerkat vocalizations. This small-scale study was designed to test two hypotheses: (a) that our domain-specific Bird-MAE provides transferable benefits to other fine-grained bioacoustic tasks, even when used as a frozen feature extractor, and (b) that prototypical probing remains superior to linear probing in this new domain.

**Dataset and preprocessing.** The MeerKAT dataset consists of 10-second multi-label audio clips with eleven unique call types originally sampled at 8kHz. As the dataset does not provide an official train-test split,

we follow the protocol from Schäfer-Zimmermann et al. (2024) by randomly sampling 20% of the data to create a test dataset. To ensure efficient experimentation while demonstrating relative performance gains, we randomly sample 50% of the remaining data for our training set with a small validation split. During data loading, all audio recordings were upsampled to to match the pretraining specifications of our models. For compatibility with Bird-MAE's 5-second input window, the 10-second clips from the `MeerKAT` dataset were segmented into two non-overlapping 5-second chunks.

**Training and augmentation.** For this study, we evaluate frozen representations from both the general-purpose Audio-MAE and our domain-specific Bird-MAE. We apply linear probing and our proposed prototypical probing to both models. The hyperparameters for the probing heads (e.g., learning rate, number of prototypes) are kept consistent with those used for the main experiments. We use a minimal augmentation strategy, applying only multi-label mixup with a probability of 0.5 during training.

