# OpenReview forum: "Can Masked Autoencoders Also Listen to Birds?"
_TMLR — Accepted by TMLR_

### Review · Reviewer_RnKm · 2025-06-24

**Summary Of Contributions:**

This paper investigates the effectiveness of self-supervised learning for bird classification tasks. The contributions of the work include: 1)
adapting the MAE pipeline for bird sound classification; 2) proposing prototypical probing to enhance the utility of frozen MAE representations in bird sound classification; 3) establishing a novel few-shot multi-label classification benchmark. The paper achieves SOTA results on the BirdSet and the performance is much better than the supervised baseline.

Although SSL has been previously studied in bird classification, the paper has thoroughly investigated the potential in SSL and largely improved the baselines based on the results.

**Audience:**

Yes

**Claims And Evidence:**

Yes

**Requested Changes:**

I find the paper is well written. I don't see any major issues.

**Strengths And Weaknesses:**

The strengths of the paper are:

1. SOTA results compared to previous supervised and SSL methods.
2. Systematic study of using SSL for bird classification, including pre-training, finetuning and representation probing.
3. Novel prototypical probing and establishing few-shot benchmark for bird classification

I find the paper is well written. I don't find obvious weaknesses for the paper.

---

> ### Author Response · Authors · 2025-07-03
> **Response to Reviewer RnKm**
>
> Dear reviewer [RnKm],
>
> We thank you for your time and effort for reviewing our paper. We are very happy for your very positive assessment and for recognizing the core contributions of our work, including the adaptation of the MAE pipeline for bird sound classification, the proposal of prototypical probing for pretrained SSL models, and the establishment of a new few-shot benchmark on BirdSet. We are particularly encouraged by your acknowledgement of our systematic study.
>
> During the rebuttal period, we have taken the opportunity to further refine the paper based on feedback from all reviewers. These revisions include: *Clarifying our contributions regarding prototypical probing*, *the added section on discussion on limitations*, and *reframing the motivation* in the introduction. We believe these changes have further strengthened the paper's narrative and clarity. We would, of course, be very happy to address any additional points or critiques you may have in the revised version.

---

### Review · Reviewer_WHLn · 2025-06-24

**Summary Of Contributions:**

* Show that Audio-MAE, even though it is trained on a wide range of diverse audio, including bird sounds, is substantially worse for bird sound classification, compared to a model which is specifically tuned for bird sounds.

* Proposes a model training pipeline specifically to train an audio MAE model which is good for classifying bird sounds. The most important addition is to apply bird-specific augmentations. All other changes also help and add up, but are minor compared to the bird-specific augmentation (even BirdSet instead of AudioSet only helps a little). This results in a new model: Bird-MAE.

* Prototypical probing is compared to linear probing on top of Bird-MAE and other models for the downstream task benchmark.

* A new type of benchmark on top of BirdSet for few-shot multi-label classification, where only very few (1, 5 or 10) training examples are used.

**Audience:**

Yes

**Broader Impact Concerns:**

-

**Claims And Evidence:**

Yes

**Requested Changes:**

Reframe the motivation and efficiency argument in the introduction. This should explicitly acknowledge the high one-time cost of pretraining Bird-MAE. I would argue, having a general base model and only needing fine-tuning would be more efficient.

Discuss about alternatives, e.g. how to improve a general audio model to better handle such niche tasks.

Rephrase the contribution/novelty on prototypical probing: Stating that this is applied here, but not giving the impression that this is totally novel. Maybe only saying that it is novel to apply this for probing specifically?

Describing more directly/exactly what the difference of prototypical probing is compared to AudioProtoPNet in Heinrich et al. (2025).

Evaluate also on some other (non-bird) test sets to see how much the methods (e.g. more epochs, mixup, etc) help there.


Some more detailed thoughts:

I would think, generic Audio-MAEs are maybe not big enough, so they only learn the most common present information. Or maybe the used SSL method is not good in catching features which are not so common in the data such as bird sounds, and other SSL methods could be better? This is maybe beyond the scope of this work to study this, but at least this should be discussed.

Intro "Adapting a general-purpose model to new domains via fine-tuning is resource-intensive, requiring high computational resources and extensive labeled data": Yes, but training a new model from scratch is even more expensive? Or you mean it is better if the general-purpose model does not need fine-tuning but it's already good enough as-is? (That's what you mean by "Leveraging frozen representations with lightweight probes"?) Sure. But then you go on and argue "...  it calls for a domain-aware adaptation of the entire SSL pipeline, including pretraining, fine-tuning, and frozen feature utilization." - so this is even much much more expensive. So I don't really understand the argument/motivation about efficiency, when the proposed method is doing exactly the opposite? If efficiency would be the goal, I would study how to make the general-purpose model (or SSL method) good enough / better to also handle such niche domains better. Not just to say that one specific general-purpose model can't do it well (then incorrectly generalizing from that that all general-purpose models can't do that well), and saying it needs domain-specific SSL.



Other random comments:


I don't know what this %_p notation means. I have never seen that. I would avoid it, as this just adds confusion. Or at least explain it somewhere, and give some reference where this comes from.

MAP is a non-standard abbreviation. While reading the abstract, I was thinking about maximum a posteriori (MAP), but the sentence did not really make sense to me. It seems you mean "mean average precision" (MAP). This abbreviation should not be used in the abstract, or at least not without explaining it.


In speech recognition, SpecAugment is an extremely helpful augmentation. Was this tried? Would this help here?




PS: I found it funny that in your Slurm train script, you write "Error: srun failed. Sleeping for 2 hours...", but then you sleep for 6h.

PPS: I saw JEPA in your code. You don't write anything about that. I guess you tried it but it does not work yet? Maybe you can add some comments in the appendix about "some preliminary JEPA experiments"?

**Strengths And Weaknesses:**

Strengths:

* Training code open, inference code open, model open.

* Comprehensive experiments, very detailed ablations (I very much like Figure 3).

* Good improvements specifically for the studied downstream task.


Weaknesses:

* The introduction argues against the high cost of fine-tuning general-purpose models but then proposes a solution  (building a domain-specific model from scratch) that is vastly more resource-intensive. This creates a logical inconsistency in the paper's motivation. The "efficiency" of their final solution lies at the inference and downstream adaptation stage (via cheap probing), but this comes at the cost of a massive up-front pretraining investment.

* The paper shows that Audio-MAE fails on this fine-grained task but doesn't deeply discuss why. It jumps to the conclusion that domain-specific SSL is the necessary solution without exploring other hypotheses. Perhaps the Audio-MAE model is simply too small, and having a larger generic audio model would not have this problem. Or the MAE reconstruction task itself is less suited for learning sparse, fine-grained features like bird calls from a general dataset compared to other SSL methods (e.g., contrastive learning, or BYOL).

* Novelty of prototypical probing: It seems that Heinrich et al. (2025) in AudioProtoPNet proposed a very similar method. This is claimed as one of the core novel contributions here, but it's not really novel.

* Does not study how much the suggested changes (e.g. more epochs, mixup, etc) help on other downstream tasks from AudioSet.

---

> ### Author Response · Authors · 2025-07-03
> **Response to Reviewer WHLn: (W1) Reframing the motivation (1/5)**
>
> Dear reviewer [WHLn],
>
> We thank you for your comprehensive feedback. It helped us identify some areas where we can improve the clarity, framing, and discussion in the paper. We are happy that you recognized the paper’s strength: open-sourced code and models, comprehensive experiments, and strong downstream gains. We have responded to every point in detail below and revised the paper accordingly. All changes prompted by a your comments are marked in [BLUE], edits addressing overlapping issues raised by multiple reviewers appear in [ORANGE]. We believe these revisions substantially strengthen the work while we preserve the paper’s main contributions and core message. We appreciate the opportunity to refine it.
>
> **(W1) Reframing the motivation**
>
> > The introduction argues against the high cost of fine-tuning general-purpose models but then proposes a solution (building a domain-specific model from scratch) that is vastly more resource-intensive. This creates a logical inconsistency in the paper's motivation [...]
>
> > Reframe the motivation and efficiency argument in the introduction. This should explicitly acknowledge the high one-time cost of pretraining Bird-MAE. I would argue, having a general base model and only needing fine-tuning would be more efficient.
>
> >Intro "Adapting a general-purpose model to new domains via fine-tuning is resource-intensive, requiring high computational resources and extensive labeled data": Yes, but training a new model from scratch is even more expensive? Or you mean it is better if the general-purpose model does not need fine-tuning but it's already good enough as-is? [...]
>
> **Summary and explanation**
>
> Thanks for pointing this out! We agree that our initial framing can create a logical inconsistency by mixing the high cost of fine-tuning with our proposed solution, which involves the even greater upfront cost of pretraining a new model. We think that this was not articulated clearly enough on our end.
>
> **Detailed revisions utilizing your feedback**
>
> We have revised the “Introduction” and added a new Section “Discussion and Limitations” sections to reframe our argument around the strategic trade-offs of creating a domain-specific foundation model. Our revised motivation better points out our initial goal: that our approach is not cheaper upfront, but rather that it represents a necessary investment to unlock high niche-performance and downstream efficiency for various tasks in challenging domains where general-purpose models seem to fall short. Our revised paper now clarifies the following points to address the concerns raised:
>
> 1. *Performance ceiling of general models (Introduction)* We now emphasize more strongly that for difficult, fine-grained tasks (e.g., BirdSet), existing general-purpose models like Audio-MAE fail to reach SOTA  performance, even with full fine-tuning (as demonstrated by our results in Table 6 and 7). This establishes the need for an alternative approach if the goal is to maximize performance in this specialized domain.
> 2. *Reframing “efficiency” to focus on the downstream lifecycle (Introduction & Discussions/Limitations):* We have adjusted the narrative to clarify that the primary efficiency benefit is in the *downstream adaptation and deployment stages*. We argue that the upfront cost of pretraining Bird-MAE is a *one-time investment (like you said)* to create a powerful, reusable foundation model for the bioacoustics community:
>     - *Efficient adaptation:* Researchers can adapt Bird-MAE to new sub-tasks with minimal data and compute by training only a probe on the frozen model. This is vastly more efficient for the users than attempting to fully fine-tune a (probably) less effective general-purpose model for each new use case. This is especially important for uses like ecologists who may have limited compute resources for retraining (or deployment on edge devices).
>     - *Data efficiency:* The strong few-shot performance demonstrates that effective models can be created from a handful of labeled examples, a critical advantage in a domain where data for rare species is scarce.
>
> We believe this revised framing more accurately and transparently presents the motivation and implications of our work.

---

> ### Author Response · Authors · 2025-07-03
> **Response to Reviewer WHLn: (W2) Why Audio-MAE fails & alternatives (2/5)**
>
> **(W2) Why Audio-MAE fails & alternatives**
>
> > The paper shows that Audio-MAE fails on this fine-grained task but doesn't deeply discuss why. It jumps to the conclusion that domain-specific SSL is the necessary solution without exploring other hypotheses. [...]
>
> > I would think, generic Audio-MAEs are maybe not big enough, so they only learn the most common present information. Or maybe the used SSL method is not good in catching features which are not so common in the data such as bird sounds [...]
>
> **Summary and explanation.**
> We thank you for this thoughtful critique and suggesting alternative explanations for why general-purpose models may underperform on fine-grained tasks. We agree that a deeper discussion strengthens the paper and clarifies the contribution. While a full exploration of how to improve general-purpose models for all niches or deeply analyzing their intrinsical problems is a very interesting research area, we agree with you that it is beyond the scope of this work. Our paper's core contribution is to demonstrate that a domain-specific adaptation of the MAE pipeline yields SOTA  results. We argue that neither of the two alternative hypotheses fully accounts for our results (more details below).
>
> **Detailed revisions utilizing your feedback.**
> In response to your feedback, we have made several revisions to better contextualize our approach and discuss alternatives:
>
> 1. *Strengthening the "Why" (Introduction):* We now more prominently feature the argument that bird sound classification is a challenging fine-grained task characterized by *low inter-class variation* and *high intra-class variation*. We state that a model pretrained on the coarse-grained, high inter-class variation of a general dataset like AudioSet is not inherently optimized for these specific fine-grained challenges. This can be seen in related work such as [Ghani et al. 2023: Global birdsong embeddings enable superior transfer learning for bioacoustic classification] or the HEAR benchmark.
> 2. *Adding a discussion of alternative hypotheses (Discussion and Limitations)*: In the new section, we directly address the alternative hypotheses raised, incorporating evidence from our own experiments and the literature:
>     1. **On the SSL Method:**  We argue that the performance gap is likely not due to the MAE reconstruction task alone. We now discuss results from related work showing that other models also struggle on bioacoustic benchmarks like BIRB (e.g., YamNET). Furthermore, we point to our own results in the benchmark, where other domain-adapted SSL models (contrastive SimCLR and masking-based AVES) are outperformed by our Bird-MAE, suggesting that the comprehensive pipeline adaptation is a dominant factor for success in this domain.
>     2. **On Model Size:** We discuss the hypothesis that a larger general-purpose model might suffice. We provide counterpoints by noting that: (a) even very large general audio models have been shown to underperform on simple bioacoustic tasks in benchmarks like HEAR; (b) our own experiments with Bird-MAE-H showed diminishing returns compared to Bird-MAE-L, suggesting size is not the only solution.

---

> ### Author Response · Authors · 2025-07-03
> **Response to Reviewer WHLn: (W3) Prototypical Probing & AudioProtoPNet (3/5)**
>
> **(W3) Prototypical Probing & AudioProtoPNet**
>
> > Novelty of prototypical probing: It seems that Heinrich et al. (2025) in AudioProtoPNet proposed a very similar method. This is claimed as one of the core novel contributions here, but it's not really novel.
>
> > Rephrase the contribution/novelty on prototypical probing [...]
>
> **Summary and explanation.** This is a very crucial point regarding the novelty of prototypical probing and its relationship to prior work, this was also mentioned by reviewer [WHLn]. We agree that our initial phrasing overstated the novelty of the mechanism itself. We have revised the paper to more precisely define our contributions.
>
> **Detailed revisions utilizing your feedback.** To adress your point, we have made the following revisions throughout the paper, specifically in the introduction, related work and methods sections:
>
> 1. *Clarifying the core novelty of prototypical probing (Introduction/Contribution)*: We have reframed our contribution. Prototypical probing is now explicitely defined as the specific application and systematic evaluation of a prototype-based head as a lightweight, parameter-efficient probing mechanism for leveraging frozen representations in SSL models. Prior work (e.g., AudioProtoPNet) showed prototype layers in fully supervised, many-class settings, our focus is their effectiveness as a lightweight probe (also in a few-shot setting): an area where MAE features are notoriously bad.
> 2. *Evaluation context (Introduction/Contribution)*: We emphasize that an important contribution is the comprehensive comparison against against a suite of established probing methods (linear, MLP, attentive) in different data regimes.
>
> These changes properly acknowledge foundational prototype-based research while positioning our work as a novel, systematic study of prototype heads for probing challenging MAE representations.

---

> ### Author Response · Authors · 2025-07-03
> **Response to Reviewer WHLn: (W4) Other non-bird downstream tasks (4/5)**
>
> **(W4) Other non-bird downstream tasks**
>
> > Does not study how much the suggested changes (e.g. more epochs, mixup, etc) help on other downstream tasks from AudioSet.
>
> > Evaluate also on some other (non-bird) test sets to see how much the methods (e.g. more epochs, mixup, etc) help there.
>
> **Summary and explanation.**
> Thank you for this suggestion. Evaluating the generalizability of our entire adapted pretrained recipe (e.g., extended epochs, mixup etc.) to non-bird audio tasks is an interesting research question to follow. However, we believe a full empirical validation is beyond the paper’s scope for two main reasons:
>
> 1. *Domain-Specific Focus*: Our story centers on holistic adaptation to a challenging, fine-grained domain (bird sound classification). Our modifications, including extensive hyperparameter/modules tuning for the pretraining recipe, were specifically optimized for the properties of bird sounds. Applying this bird-tuned recipe unchanged to speech or general audio could mix domain effects with recipe effects and obscure our claims.
> 2. *Experimental overhead*: A new investigtation of different domains as you suggest would require one of the two extensive setups:
>     1. *New domain replication*: Re-running the entire pipeline for each new domain would require curating large, task-matched pre-training sets and training fresh MAEs, an effort comparable to a new paper.
>     2. *AudioSet re-pretraining:* This would involve taking our recipe modifications (e.g. 150 epochs etc.) and applying them to a new full pretraining on AudioSet itself (that has already been tuned/optimized in the original paper, longer training and mixup in pretraining did not improve model performance), followed by evaluation on its downstream task. This also represents a large-scale pretraining effort that is separate from our primary contribution of creating a domain-specific model.
>
> **Detailed revisions utilizing your feedback**
> 1. *Framing in future work (Discussions and Limitations)* : In the revised section, we now explicitly state that evaluating the transferability of the full suite of pretraining recipe optimizations to general audio benchmarks like the full AudioSet is an important direction for future research.
> 2. *New experiment on general audio (Discussion and Limitations):* To further test the transferability of our most successful component, protoypical probing, we have conducted a new preliminary experiment. We now also present results for prototypical probing to the frozen Audio-MAE and evaluate it on the general audio benchmark AS-20k.
> 3. *New experiment on a bioacoustic task (Discussion and Limitations):* To test the generalizability beyond bird sounds, we have conducted a *new set of preliminary experiments* on a different fine-grained bioacoustic domain: the MeerKAT mammal vocalization dataset [Schäfer-Zimmermann et al. 2024: animal2vec and MeerKAT: A self-supervised transformer for rare-event raw audio input and a large-scale reference dataset for bioacoustics]. This allows us to test two key hypotheses: (a) if our prototypical probing remains superior to linear probing, and (b) if our domain-adapted Bird-MAE offers any cross-species benefit over the general-purpose Audio-MAE.
>
> These updates clarify why a full multi-domain study is out-of-scope while still showing that our innovations can extend beyond bird audio. If further clarification would help, we are happy to elaborate.

---

> ### Author Response · Authors · 2025-07-03
> **Response to Reviewer WHLn: Other random comments (5/5)**
>
> **Other random comments**
>
> > I don't know what this %_p notation means. […]
>
> Thanks for pointing this out, we get why this can be confusing. We have revised the paper to use the standard “pp” (percantage points) notation when referring to the absolute difference in mAP scores and "%" when referring to the mAP score value itself, to improve clarity and adherence to common practice.
>
> > MAP is a non-standard abbreviation. While reading the abstract, I was thinking about maximum a posteriori (MAP), but the sentence did not really make sense to me. It seems you mean "mean average precision" (MAP). This abbreviation should not be used in the abstract, or at least not without explaining it.
>
> We agree that this can be confusing. In the revised absract and at its first use n the main text, we explicitely it as “Mean Average Precision” and change the abbreviation to “mAP” (e.g., like in the original Audio-MAE paper) to avoid any ambiguity.
>
> > In speech recognition, SpecAugment is an extremely helpful augmentation. Was this tried? Would this help here?
>
> Our current spectrogram augmentation pipeline (frequency and time masking, detailed in Appendix D.1) is indeed inspired by and very similar in principle to SpecAugment. As far as I understand, we are functionally doing SpecAugment minus the time-warping. We found this combination effective within our broader augmentation strategy, adding time-warping did not seem to improve performance. What was a performance game changer was especially the mixup directly on the waveforms like we describe in D.1.
>
> > I found it funny that in your Slurm train script, you write "Error: srun failed. Sleeping for 2 hours...", but then you sleep for 6h.
>
> I'm genuinely very surprised someone notices some code artifacts (for whatever reason this was done)! That line is of course our secret test to see who really checks the code in detail.
>
> > I saw JEPA in your code. You don't write anything about that. I guess you tried it but it does not work yet? Maybe you can add some comments in the appendix about "some preliminary JEPA experiments"?
>
> This was indeed part of our broader exploratory work during the experiments. However, as these experiments are preliminary and not yet mature enough for inclusion, we focused the current paper on the MAE findings which yielded notable and validated results. We agree that exploring JEPA for this domain is an interesting future direction but we would rather exclude it from the paper for now.

---

### Review · Reviewer_XGfz · 2025-06-29

**Summary Of Contributions:**

This paper presents Bird-MAE, a domain-adapted masked autoencoder framework for bird sound classification. Recognizing the limitations of general-purpose audio MAEs (e.g., Audio-MAE trained on AudioSet), the authors propose a holistic adaptation across three modules: (M1) pretraining on curated BirdSet data with optimized SSL recipe, (M2) fine-tuning with domain-specific augmentations and a novel prototypical pooling, and (M3) a parameter-efficient prototypical probing method that enhances the use of frozen features. The model achieves state-of-the-art (SOTA) performance on the BirdSet multi-label benchmark and demonstrates strong few-shot learning capability. Notably, prototypical probing narrows the performance gap to full fine-tuning to ~3%p, outperforming linear probing by up to 37%p in MAP.

**Audience:**

Yes

**Broader Impact Concerns:**

I do not have major ethical concerns with this submission. The authors focus on improving bird sound classification through domain-adapted self-supervised learning, using publicly available datasets and releasing code/models to the community. These efforts can support positive use cases in ecology, biodiversity monitoring, and conservation, especially given the model's strong few-shot capabilities which could help in low-resource regions or with rare species.

That said, as with any audio recognition model trained at scale, it is worth noting that deployment in real-world monitoring systems (e.g., in forests, reserves, or urban spaces) raises general concerns about surveillance, unintended data capture, and responsible use. While not specific to this paper, future applications should involve clear usage guidelines and transparency for communities affected by automated sensing systems.

Overall, the paper adheres to standard ethical practices, and the broader impact is likely to be positive, particularly in the context of open science and ecological research.

**Claims And Evidence:**

Yes

**Requested Changes:**

1. Clarify what’s truly novel in prototypical probing. The proposed probing method is effective, but it builds directly on prior work (e.g., prototypical networks, class token readouts, and attention-based pooling). It would help if the authors could more clearly articulate what makes their version different — is it the spatial max-pooling? the constrained linear projection? the choice of cosine similarity? Right now, it’s not obvious whether this is a new method or a clever reuse of existing pieces.

2. Discuss generalization beyond birds. The paper focuses entirely on bird vocalizations, which is reasonable for a domain-specific model — but it would still be helpful to add some discussion (or preliminary experiments) on whether the benefits of prototypical probing or the Bird-MAE setup could carry over to other fine-grained audio tasks (e.g., dialect classification, insect calls, or low-resource speech). Even a negative result would be informative.

3. Expand prototype interpretability analysis. Figure 5 is a nice start, but it’s quite limited. Since this paper touches on a domain (bioacoustics) where interpretability can actually matter — e.g., for ecologists or conservationists — it would be worth digging a little deeper. Are prototypes mapping onto specific call types? Harmonic structures? Species clusters? Even one or two examples would enrich the narrative.

**Strengths And Weaknesses:**

This paper tackles the domain gap in applying general-purpose self-supervised audio models, such as Audio-MAE, to fine-grained tasks like bird sound classification. The authors propose Bird-MAE, a domain-specific masked autoencoder pretrained on a curated subset of BirdSet, and introduce prototypical probing as an efficient method to use frozen MAE representations. Across eight BirdSet benchmarks, the proposed approach achieves strong gains over both supervised baselines and previous self-supervised models. The improvement from frozen features using prototypical probing is particularly noteworthy, outperforming linear probing by a large margin and narrowing the gap to full fine-tuning to just a few points in MAP. The authors also provide thorough ablations on pretraining data, masking, pooling methods, and probing strategies, which gives the paper a strong empirical backbone. The release of code, pretrained models, and the new few-shot benchmark further enhances the paper’s value to the community.

Where the paper is slightly weaker is in terms of technical novelty. The backbone architecture (ViT-MAE) is taken directly from prior work, and while the prototypical probing idea is well-executed, it draws heavily from existing ideas in prototypical networks and attentive pooling. The probing layer is essentially a non-parametric readout module with class-wise centroids, and while the efficiency and effectiveness are appreciated, the novelty is modest. Additionally, the evaluation is focused solely on bird sound data — which makes sense given the paper’s goal — but leaves questions about how transferable the insights or methods are to other domains, such as speech, environmental sounds, or multi-modal settings.

Overall, this is a strong applied paper that clearly demonstrates how to adapt self-supervised learning pipelines to challenging, fine-grained audio domains. While it doesn’t introduce new core learning principles, it’s well-motivated, carefully designed, and thoroughly evaluated, with takeaways that are likely to be useful for others working in low-resource or bioacoustic settings.

---

> ### Author Response · Authors · 2025-07-03
> **Response to Reviewer XGfz: (W1) Technical novelty and clarifications (1/3)**
>
> Dear reviewer [XGfz],
>
> Thank you for your detailed and thoughtful feedback on our paper. We are very encouraged that you found the strengths of the paper as a strong applied paper with a strong empirical backbone, and appreciating the SOTA results, ablations and contributions for open sience, especially in the niche domain. We agree with your assessment regarding areas for improvement. We adress all your concerns in the following and think that we can improve the paper while still preserving the paper’s main contributions and core message. All changes prompted by a your comments are marked in [RED], edits addressing overlapping issues raised by multiple reviewers appear in [ORANGE].
>
> **(W1) Technical novelty and clarifications**
>
> > The backbone architecture (ViT-MAE) is taken directly from prior work, and while the prototypical probing idea is well-executed, it draws heavily from existing ideas in prototypical networks and attentive pooling. The probing layer is essentially a non-parametric readout module with class-wise centroids, and while the efficiency and effectiveness are appreciated, the novelty is modest.
>
> > Clarify what’s truly novel in prototypical probing. The proposed probing method is effective, but it builds directly on prior work (e.g., prototypical networks, class token readouts, and attention-based pooling). It would help [...]
>
> **Summary and explanation.**
> Thank you for the feedback on technical novelty. This was also mentioned by reviewer [WHLn]. We agree that our initial presentations did not distinguish our proposed technique from its inspirations sufficiently. Our primary novelty is not the invention of completely new prototype-based layers, but rather their specific application and systematic evaluation as a lightweight probing mechanism in a new context: for frozen SSL (here MAE) representations. We would also like to clarify a point: while our method has similarities to prototypical networks, it is in fact a parametric approach. Your statement would be true for classic prototypical networks the compute fixed centroids from training examples. In contrast, our methods learn a fixed set of trainable parameters during the probing stage: (1) the prototype vectors themselves, optimized via backprop, and (2) the weights of the final linear layer that maps the similarity scores to a single logit for each class (explained in e.g., page 5 “prototypical pooling”). We believe this parametric nature is a key reason for its effectiveness, as it allows the model to learn abstract representations of class characteristics rather than being restricted to the simple mean of training exemplars.
>
> **Detailed revisions utilizing your feedback.**
> To address your comment, we have made the following revisions throughout the paper, primarily in the Introduction, Related work and Methods sections:
>
> 1. *Reframing the contribution (Introduction)*: We have revised our framing to explicitly state that our core novelty is the specific application and evaluation of a prototype-based head as a lightweight, parameter-efficient probing mechanism for frozen representations. We clarify that while prior work like AudioProtoPNet has shown the effectiveness of such heads in fully supervised settings, their use and strong performance as a *probe* for frozen SSL features, particularly from MAEs, which are notoriously challenging to probe effectively, is a key finding and contribution of our work. Additionally, they have not yet been used in a few-shot setting.
> 2. *Evaluation context (Introduction/Contributions)*: We now emphasize that a notable part of our contribution lies in the novel usage and **systematic comparison** of prototypical probing against a suite of established probing methods (linear, MLP, and attentive) on MAE (and other) features (in a multi-label and few-shot setting). This rigorous evaluation, which demonstrates its superior performance and parameter efficiency, is a novel aspect of our study.
>
> We believe these revisions now better credit the foundational work while accurately situating our novel contributions.

---

> ### Author Response · Authors · 2025-07-03
> **Response to Reviewer XGfz: (W2) Generalization beyond birds (2/3)**
>
> **W2: Generalization beyond birds**
> > Additionally, the evaluation is focused solely on bird sound data — which makes sense given the paper’s goal — but leaves questions about how transferable the insights or methods are to other domains [...]
>
> > Discuss generalization beyond birds. The paper focuses entirely on bird vocalizations, which is reasonable for a domain-specific model — but it would still be helpful to add some discussion (or preliminary experiments) on whether the benefits of prototypical probing or the Bird-MAE setup could carry over to other fine-grained audio tasks [...]
>
> **Summary and explanation.** Thank you for this excellent suggestion to discuss and evaluate the generalizability of our methods beyond bird sounds. We agree that this is a crucial point for assessing the broader impact of our work. While a comprehensive evaluation across multiple new domains would be beyond the scope of this paper (as you also state), we have followed your suggestions to provide initial evidence and expand our discussion on the topic.
>
> **Detailed revisions utilizing your feedback.**
> To utilize your feedback, we have made the following revisions:
>
> 1. *Preliminary experiment on general audio (Discussion and Limitations):* To further test the transferability of our most successful component, prototypical probing, we have conducted a new preliminary experiment. We now also present results for prototypical probing to the frozen Audio-MAE and evaluate it on the general audio benchmark AS-20k in the new section.
> 2. *Preliminary experiment on meerkats (Discussion and Limitations):*  To test the generalizability beyond bird sounds, we have conducted a new set of preliminary experiments on a different fine-grained bioacoustic task: the MeerKAT mammal vocalization dataset [Schäfer-Zimmermann et al. 2024: animal2vec and MeerKAT: A self-supervised transformer for rare-event raw audio input and a large-scale reference dataset for bioacoustics]. This allows us to test two key hypotheses: (a) if our prototypical probing remains superior to linear probing, and (b) if our domain-specific Bird-MAE offers any cross-species benefit over the general-purpose Audio-MAE.
> 3. *Expanding discussion on generalizability (Discussion and Limitations):* We now explicitly state that exploring the generalizability of prototypical pooling and probing to other fine-grained audio tasks (e.g., insect calls, mammal sounds) and even other modalities (vision, text) is a crucial next step.
>
> We believe these additions directly address your request.  By providing concrete preliminary experimental evidence of generalizability on non-bird tasks, we strengthen the paper's claims and better frame the exciting potential for future cross-domain research.

---

> ### Author Response · Authors · 2025-07-03
> **Response to Reviewer XGfz: (W3) Expanding prototype interpretability analysis (3/3)**
>
> **(W3) Expanding prototype interpretability analysis**
>
> > Expand prototype interpretability analysis. Figure 5 is a nice start, but it’s quite limited. Since this paper touches on a domain (bioacoustics) where interpretability can actually matter — e.g., for ecologists or conservationists — it would be worth digging a little deeper [...]
>
> Thank you for this suggestion. We fully agree that a deep dive into prototype interpretability is a valuable research direction, especially given the possible practical applications in bioacoustics. While we are interested about this direction, we have chosen to keep the interpretability analysis in the current paper concise for the following reasons:
>
> - *To maintain a clear focus on our core contribution:* demonstrating the surprising and very notable *performance benefits* of using a prototype-based layer for probing and fine-tuning of frozen MAE features. This is a side effect not typically the primary topic in works that primarily focus on the explainability of such methods [Heinrich et al. 2024: AudioProtoPNet].
>
> - *Scope of a comprehensive study:* We think that a comprehensive interpretability study would be a very substantial research direction in itself (how to evaluate, what bird vocalization types, what training regimes: fine-tuning vs probing etc.). Given the short rebuttal time, it would rather need its own dedicated study.
>
> - *Existing work*: A detailed analysis of prototype interpretability in the context of avian bioacoustics has been thoroughly conducted in the AudioProtoPNet paper, where we draw inspiration from. While our model backbones and training paradigms differ, their work provides an excellent and comprehensive reference for how such prototypes can be visualized and interpreted in this domain.
>
> **Detailed revisions utilizing your feedback.**
> To acknowledge the importance of this topic and provide more context for the reader without disrupting the paper's focus, we have made the following revision:
>
> 1. *Extended prototype discussion** (Discussion and Limitations): We have expanded our mention of interpretability. We now more explicitly reference the detailed analysis available in [Heinrich et al. 2024: AudioProtoPNet] as a prime example of what these prototypes can reveal (e.g., mapping to distinct call types). We then frame a similar deep dive for our Bird-MAE model and its learned representations as a promising direction for future work.
>
> We believe this approach respectfully addresses your suggestion by enriching our discussion and pointing readers to relevant in-depth work, while maintaining the focused performance-oriented narrative of the current paper.

---

### Author Response · Authors · 2025-07-14
**Authors' General Comment and Summary**

Dear reviewers,

We want to thank you again for the detailed and constructive feedback, and for recognizing the paper’s following strengths:

- **SOTA performance on BirdSet and strong few-shot capability** *[XGfz WHLn, RnKm]***:** The Bird-MAE model notably outperforms prior supervised and SSL baselines, delivering SOTA and narrowing the gap between probing and full fine-tuning.
- **Thorough empirical study with extensive ablations and benchmarks** *[XGfz, WHLn]***:** The paper provides a  systematic study of SSL adaptation across pre-training, fine-tuning and probing stage, giving the work a solid empirical backbone.
- **Prototypical probing to utilize frozen (MAE) features** *[XGfz, RnKm]*: The proposed probe markedly outperforms linear probing while remaining parameter-efficient, demonstrating a practical path to deploy frozen SSL representations.
- **Open science contributions** *[WHLn]:* Releasing training/inference code, Bird-MAE weights, and the novel few-shot BirdSet benchmark makes the work readily reproducible and useful for the community.

We have incorporated all major suggestions and highlighted changes in [RED] for reviewer [XGfz], [BLUE] for reviewer [WHLn], and [ORANGE] for overlaps throughout the manuscript. The most important revisions are:

- **Clarified novelty** *[WHLn, XGfz]:* We reframed prototypical probing as a *probe-level* contribution for frozen (MAE/SSL) features, now more clearly positioning it relative to AudioProtoPNet.
- **Reframed motivation & efficiency** *[WHLn]:* We reframed the Introduction and added a new *Discussion & Limitations* section to explain the one-time cost of pre-training Bird-MAE and the downstream efficiency gains it unlocks.
- **Generalization evidence** *[WHLn, XGfz]:* We added additional preliminary results on (a) the AS-20k general-audio benchmark and (b) the MeerKAT mammal-call dataset, confirming that prototypical probing remains superior to linear probes and that Bird-MAE retains cross-species value for bioacoustics.

We believe these updates strengthen the paper while preserving its core message: tailoring every stage of a SSL pipeline converts frozen SSL features into a SOTA, few-shot-friendly bird sound classifier, showing that domain-aware adaptation beats general-purpose audio MAEs for fine-grained bioacoustics. We hope these revisions demonstrate our commitment to addressing the feedback thoughtfully. If you have any further questions, we would be happy to discuss them.

---

### Decision · Action_Editor_VamK · 2025-08-12

**Recommendation:** Accept with minor revision

**Additional Comments:**

Expected changes for the minor revision:

* Please define "pp" (e.g. in a footnote) as some readers may not be familiar with this abbreviation.

* SpecAugment should likely be discussed and cited in the paper given the authors acknowledge the similarity and inspiration taken from it.

**Audience:**

Yes

**Audience Explanation:**

The adaptations in the paper for the bird sound classification task achieve state-of-the-art performance and would be of interest to researchers in that field, and possibly those in the broader area of audio classification given the promising results beyond bird sounds.

**Claims And Evidence:**

Yes

**Claims Explanation:**

The paper claims state-of-the-art performance on a bird sound classification task achieved through a set of domain-specific adaptations to a standard self-supervised pipeline based on masked autoencoders. All reviewers agreed that the paper included extensive experiments, including ablation studies on various stages of the pipeline, to support the claims. Initial concerns about novelty of prototypical probing and generality of the pipeline to other tasks were addressed in the revision.